# Thioester-mediated RNA aminoacylation and peptidyl-RNA synthesis in water

Jyoti Singh[1], Benjamin Thoma[1], Daniel Whitaker[1], Max Satterly Webley[1], Yuan Yao[1] & Matthew W. Powner[1 ✉]

To orchestrate ribosomal peptide synthesis, transfer RNAs (tRNAs) must be aminoacylated, with activated amino acids, at their 2′,3′-diol moiety[1,2], and so the selective aminoacylation of RNA in water is a key challenge that must be resolved to explain the origin of protein biosynthesis. So far, there have been no chemical methods to effectively and selectively aminoacylate RNA-2′,3′-diols with the breadth of proteinogenic amino acids in water[3–5]. Here we demonstrate that (biological) aminoacyl-thiols (**1**) react selectively with RNA diols over amine nucleophiles, promoting aminoacylation over adventitious (non-coded) peptide bond formation. Broad side-chain scope is demonstrated, including Ala, Arg, Asp, Glu, Gln, Gly, His, Leu, Lys, Met, Phe, Pro, Ser and Val, and Arg aminoacylation is enhanced by unprecedented side-chain nucleophilic catalysis. Duplex formation directs chemoselective 2′,3′-aminoacylation of RNA. We demonstrate that prebiotic nitriles, *N*-carboxyanhydrides and amino acid anhydrides, as well as biological aminoacyl-adenylates, all react with thiols (including coenzymes A and M) to selectively yield aminoacyl-thiols (**1**) in water. Finally, we demonstrate that the switch from thioester to thioacid activation inverts diol/amine selectivity, promoting peptide synthesis in excellent yield. Two-step, one-pot, chemically controlled formation of peptidyl-RNA is observed in water at neutral pH. Our results indicate an important role for thiol cofactors in RNA aminoacylation before the evolution of proteinaceous synthetase enzymes.

At life's functional core, there is a complex and inseparable interplay between nucleic acids and proteins, but the origin of this relationship remains a mystery. Although nucleic acids store, replicate and transmit sequence information through their inherent structural capacity for molecular (self-)recognition[6–9], proteins are the molecular, structural and catalytic workhorses of life. Unlike nucleic acids, peptides do not innately replicate in a sequence-specific manner[10,11], so life must control and transmit the peptide sequences that are essential to its survival through nucleic acid encoding[12]. Understanding how nucleotide-controlled peptide biosynthesis could have first emerged is a notable gap in our understanding of life but is a formidable challenge owing to the immense complexity and antiquity of protein synthesis[1]. However, all proteins are built through the iteration of a universally conserved two-stage process, ribosomal peptide synthesis (RPS). In the first stage of RPS, transfer RNAs (tRNAs) are aminoacylated, loading an activated amino acid onto the 2′,3′-diol moiety of the tRNA, before these subsequently undergo ribosome-catalysed peptide synthesis (Fig. 1a). Notably, throughout RPS, the activated amino acids are orchestrated and controlled through their covalent attachment to RNA and so the selective formation of RNA esters underpins, and must predate, RPS.

RNA aminoacylation is controlled in biology by a unique family of synthetase enzymes[2]. Curiously, these enzymes implement the rules of the genetic code but are themselves synthesized by the ribosome following those same rules, making the origins of RNA aminoacylation the most intriguing causal paradox in biology. However, despite its

obvious importance, the discovery of a non-enzymatic synthesis of aminoacyl-RNA is a long-standing and hereto unresolved challenge[3–5]. RNA amino acid esters are not known to form selectively in water, posing the question of their provenance, and so we set out to develop an effective and selective RNA aminoacylation in water.

Previous studies of nucleoside aminoacylation have focused on aminoacyl phosphates **2**, aminoacyl imidazole **3** and *N*-carboxyanhydrides (NCAs; **4**)[13–16] but all of these electrophiles are highly unstable in water and difficult to prepare without substantial hydrolysis. Even more problematically, these highly activated amino acids do not react selectively with the 2′,3′-diol of RNA, as is required for tRNA aminoacylation, and so uncontrolled background amide formation is a substantial problem[15]. It is important that the formation of aminoacyl-RNAs (exploited in RPS) must occur without uncontrolled amino acid polymerization: this is a prerequisite for controlled RPS, which must outcompete random peptide synthesis to rise above the noise threshold[17]. Therefore, despite decades of research, there remains no effective means of chemically aminoacylating RNA strands terminating in a vicinal diol, with hydrolysis and polymerization occurring more readily than RNA aminoacylation[5].

Considering the poor selectivity exhibited by highly reactive electrophiles, we suspected that the solution to selective RNA aminoacylation would be found through a milder activation strategy and specifically reasoned that biological thioesters may be the ideal mode of activation to achieve selective RNA aminoacylation in water. Thioesters

[1]Department of Chemistry, University College London, London, UK. ✉e-mail: matthew.powner@ucl.ac.uk

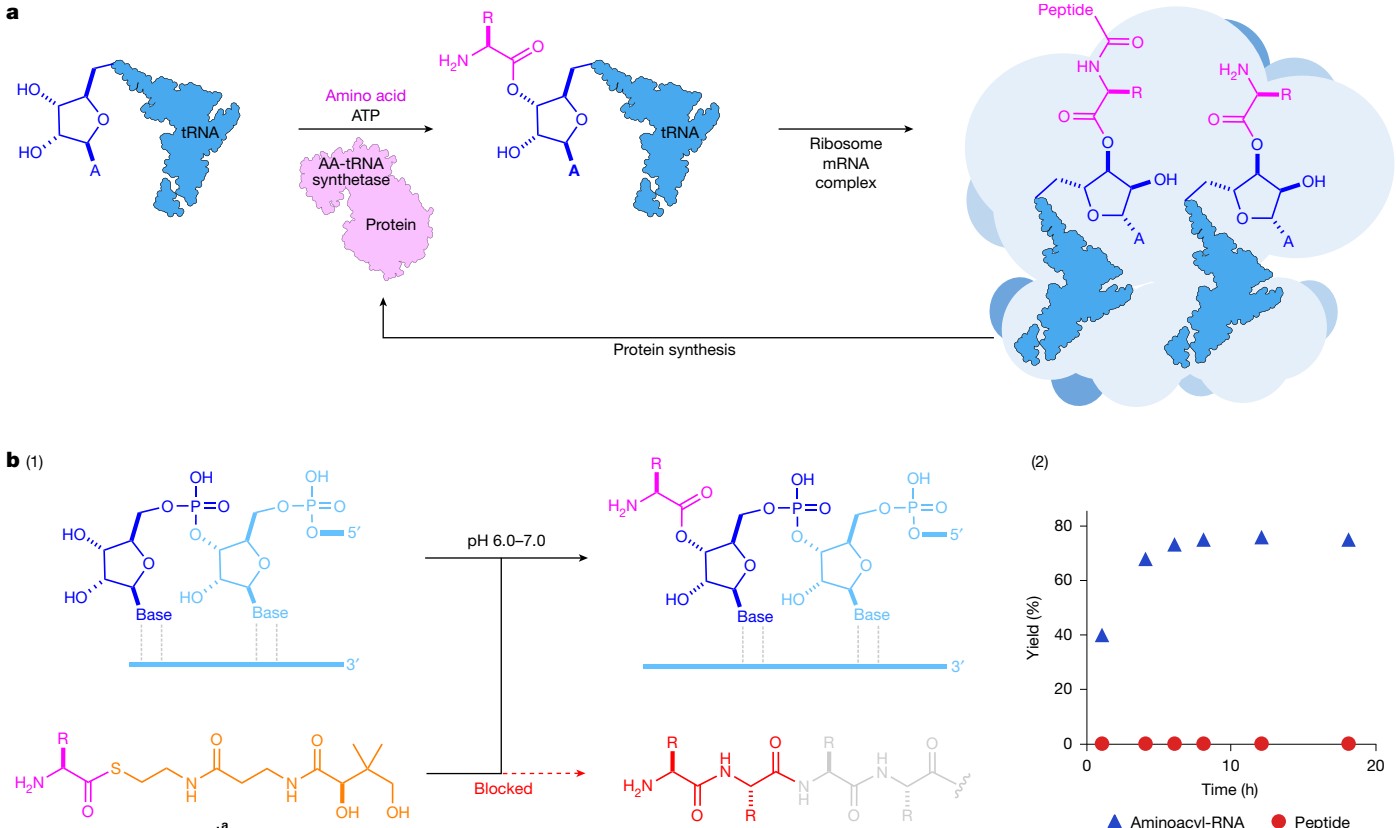

**Fig. 1 | RNA aminoacylation. a**, Schematic showing the two-step mechanism of RPS. RNA aminoacylation is controlled by aminoacyl-tRNA synthetase enzymes, which are themselves produced by RPS. This causal paradox obfuscates the origins of protein biosynthesis. **b**, (1) Highly selective non-enzymatic aminoacylation of RNA in water by biological aminoacyl-thiols ($1_{Aaa}^{a}$; Aaa = three-letter code for generic aminoacyl residue; superscript a = pantetheinyl moiety, shown in orange). The formation of aminoacyl-thiol (**1**) blocks the spontaneous formation of peptides, favouring the non-enzymatic synthesis of aminoacyl-RNAs that are required for RPS. R = amino acid side chain. Chemical structures are drawn neutral for clarity. (2) Selective and high-yielding non-enzymatic 2′,3′-aminoacylation of adenosine **16A** (2 mM) with aminoacyl-thiol $1_{Arg}^{e}$ (120 mM; superscript e = ethanethiyl moiety) in water at pH 6.5 and room temperature. Despite the large excess of **1**, and its free amine, peptide synthesis is not observed. Thioester-activated amino acids react selectively with RNA to yield aminoacyl-RNA at neutral pH.

are a crucial component of central metabolism, driving a wide range of anabolic pathways, including fatty acid, polyketide and, notably, non-ribosomal peptide syntheses[18]. Thioesters, like RNA aminoacylation, have ancient roots in biochemistry and both predate the last universal common ancestor of all life on Earth. The role of thioesters in central metabolism suggests that thioester-based metabolism (a 'thioester world') may have laid the foundations for extant life[18,19] and so we considered the role that thioesters might play in RNA aminoacylation.

The aminoacylation of RNA is just one class of biological acyl transfer that is catalysed by synthetase enzymes. However, most biological acyl-transfer reactions exploit acyl-thiols and pantetheine **5a** as a cofactor[20]. **5a**, the functional fragment of coenzyme A, is universally conserved across all organisms and is life's most prominent acyl-transfer cofactor. Although it has been proposed that aminoacyl-thiols (**1**) may hold the key to the early evolution of peptide synthesis[18], it has always previously been predicted that **1** would directly yield peptides. However, this ignores the essential and ancient role of RNA aminoacylation in protein synthesis, as well as the inherent aqueous reactivity of thioesters.

Here we demonstrate that (biological) aminoacyl-thiols (**1**) chemoselectively aminoacylate RNA at neutral pH in water. This non-enzymatic RNA aminoacylation highlights an important chemical distinction between the two separate stages of protein synthesis, that is, between RNA aminoacylation and peptide synthesis, and demonstrates how to chemically control these two steps under the same reaction conditions

in water. Aminoacyl-thiol (**1**) and (biological) thioester activation suppress adventitious peptide synthesis while delivering the activation required to achieve the first stage of RPS (Fig. 1), conceptually uniting biological thioester energy with functional RNAs to allow chemoselective diol acylation in water.

We also demonstrate that thioacid activation provides the reactivity required to orthogonally achieve the second step of RPS, that is, peptidyl-RNA synthesis, in water under the same reaction conditions. These two closely related modes of activation, thioester and thioacid activation, can direct the two chemical steps required for RPS under the same conditions in water and proceed without evolved catalysts, intramolecular or Watson–Crick templated reactivity and without purification of intermediate aminoacyl-RNAs. Together, these two reactions provide a mechanism to realize high-yielding, chemoselective, one-pot synthesis of peptidyl-RNA in water at neutral pH.

## Selective aminoacylation of nucleosides

To begin our investigation, it was first essential to determine whether ambiphilic aminoacyl-thiol **1**, which is an electrophilically activated thioester and a nucleophilic amine ($pK_{aH}$ 7.5), would spontaneously polymerize[21,22]. However, **1** was found to be very stable in water (Fig. 2b). Peptide formation from **1** was ineffective at all pHs and the main pathway was the slow hydrolysis of **1**. For example, alanine thioester L-$1_{Ala}^{e}$ reacts sluggishly at pH 7.0 to afford a very low yield of diketopiperazine

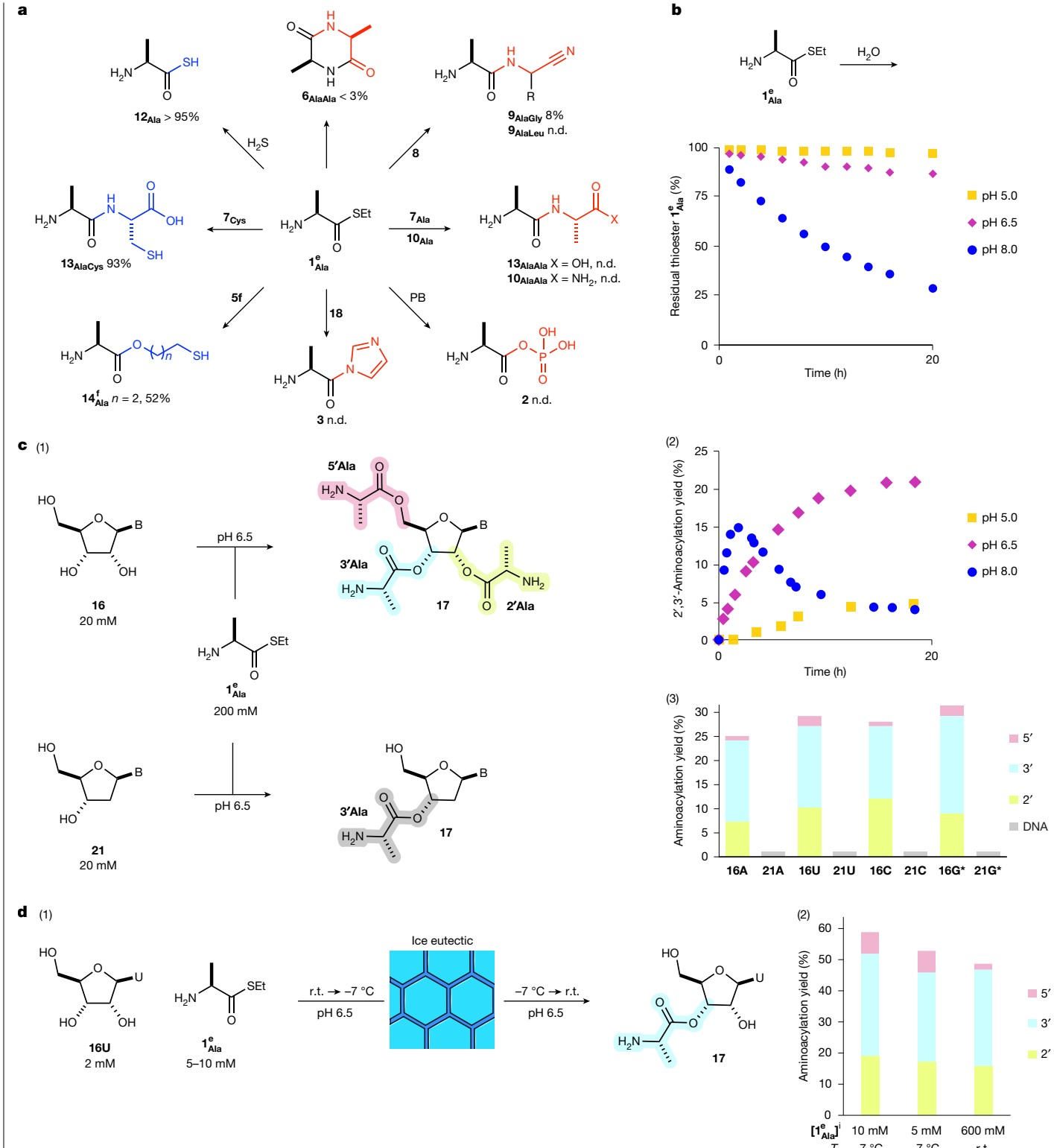

**Fig. 2 | Reactivity of nucleophiles with alanine thioester. a**, Alanine thioester L-$1^e_{Ala}$ (100 mM) reacted ineffectually with amine, imidazole (**18**) and phosphate nucleophiles (red) at neutral pH but in excellent-to-quantitative yield with sulfur nucleophiles (blue). Maximum per cent conversion with specified nucleophile (150 mM)[†] at pH 7.0 and room temperature is given. [†], 140 mM glycine nitrile (**8$_{Gly}$**); 220 mM leucine nitrile (**8$_{Leu}$**); 200 mM phosphate buffer (PB); 200 mM imidazole (**18**). n.d. = not detected. **b**, The stability of L-$1^e_{Ala}$ (200 mM) at room temperature in 2-(*N*-morpholino)ethanesulfonic acid (MES; 1 M, pH 5.0–6.5) or 3-(*N*-morpholino)propanesulfonic acid (MOPS; 1 M, pH 8.0) buffer. **c**, (1) Selective aminoacylation of ribonucleosides (**16**). (2) 2′,3′-Aminoacyl nucleoside **17** yield observed on incubating alanine thioester L-$1^e_{Ala}$ (200 mM) with adenosine **16A**

(20 mM) at room temperature in MES (1 M, pH 5.0–6.5) or MOPS (1 M, pH 8.0) buffer. See Supplementary Fig. 70 for aminoacylation yields at pH 5.0–10.0 over 100 h. (3) Alcohol-aminoacylation yield for specified nucleosides (20 mM) with L-$1^e_{Ala}$ (200 mM) at room temperature and pH 6.5 after 18 h. *2 mM **16G**; 2 mM **21G**. B = nucleobase. See Extended Data Fig. 1 for an expanded scope of nucleosides. **d**, (1) Nucleoside aminoacylation at low initial concentration. (2) Yield of uridine **16U** (2 mM) aminoacylation observed with L-$1^e_{Ala}$ (5 mM or 10 mM) when frozen, incubated at −7 °C for 5 days and then thawed at room temperature (r.t.). The yield of **17$_{Ala}$** is substantially augmented by freezing and comparable with aminoacylation of **16U** (20 mM) and L-$1^e_{Ala}$ (600 mM) at pH 6.5 and room temperature after 24 h.

(6$_{AlaAla}$, <3%), alongside alanine (7$_{Ala}$, 48%) after 5 days (Fig. 2a). More 6 was observed at higher pH but the maximum yield (21%) was observed at pH 8.0 (Supplementary Fig. 45) and hydrolysis dominated at pH 9.0–10.0 (Supplementary Fig. 74). Notably, we found that peptide formation was further suppressed at lower pHs (Fig. 2b and Supplementary Fig. 45).

Surprised by the stability of ambiphilic thioester 1, we next investigated its reaction with glycine nitrile 8$_{Gly}$ (p$K_{aH}$ 5.4), the most nucleophilic α-aminonitrile, which would be predominantly free-base amine at pH 7.0. 8$_{Gly}$ also reacted ineffectively with L-1$_{Ala}^e$, yielding dipeptide nitrile 9$_{AlaGly}$ in very low (8%) yield. Moreover, other proteinogenic aminonitriles (for example, 8$_{Leu}$), amino acids (7$_{Ala}$) and amino acid amides (10$_{Ala}$), did not react with thioester L-1$_{Ala}^e$ at all (Fig. 2a). This thioester reactivity strongly contrasts with the near-quantitative ligations observed with activated peptide thioacids (for example, Ac-(Aaa)$_n$-SH; 11) under the same, neutral pH, conditions[23,24], despite the close structural and chemical relationship between 1 and 11. This demonstrated a marked difference in the reactivity of thioesters and underlined an important principle: highly activated electrophiles react effectively with amines in water to yield peptides[24] but weakly activated thioesters do not[21]. We therefore considered that coordination of these two types of activation may hold the key to differentiating RNA aminoacylation and peptide bond formation in water under the same conditions and therefore the means to realizing effective non-enzymatic synthesis of peptidyl-RNA in water.

Surprised by the poor reactivity of aminoacyl-thiol 1 with amines, we next investigated their reaction with sulfide nucleophiles. In contrast to amines, hydrogen sulfide (H$_2$S) reacted with 1 to afford thioacids (12) in excellent-to-quantitative yield (Fig. 2a and Supplementary Table 4). Cysteine (7$_{Cys}$) also reacted with thioester 1 to yield dipeptides (13) in near-quantitative yield, by sulfide exchange and intramolecular S-to-N acyl shift. Similarly, thiopropanol 5f yielded ester 14 in good yield and this facile intramolecular esterification, by means of S-to-O acyl shift, led us to consider how aminoacyl-thiols (1) could selectively aminoacylate nucleosides at neutral pH.

We next incubated 5′-mercapto-adenosine 15 with thioester L-1$_{Ala}^e$ and observed 2′,3′-diol aminoacylation (54%), which we initially ascribed to intramolecular S-to-O acyl transfer. Notably, however, addition of cytidine (16C) as a competitor nucleophile led to intermolecular aminoacylation of 16C (Extended Data Fig. 1). Indeed, incubating 16C alone with thioester L-1$_{Ala}^e$ led to the formation of alanyl-cytidine 17$_{Ala}^C$ in good yield. Ribonucleoside-2′,3′-aminoacylation with thioester 1 was observed despite the large (about 3,000-fold) excess of water and tenfold excess of amine relative to the RNA 2′,3′-diol (Fig. 2c). This efficient and selective nucleoside aminoacylation stands in sharp contrast to the reaction of acyl-imidazoles (3), reported by Weber and Orgel, which form dipeptides more efficiently than aminoacyl-RNAs[4], and previous reports in which evolved catalysts or intramolecular tethering were required to enable RNA aminoacylation in water[25,26], demonstrating that biological thioesters are predisposed to aminoacylate RNA while suppressing peptide bond formation.

To further investigate the selectivity of aminoacyl transfer, we next challenged thioester 1 to react with phosphate and imidazole (18), which are common nucleophiles for activated carboxylates in water. We found that neither reacted with thioester 1 (Fig. 2a). Perhaps even more surprisingly, glycerol (19), which contains three alcohols and a vicinal diol, such as ribonucleosides (16), did not react with 1. This demonstrated an important but very fine balance between nucleophile and electrophile that enables ribonucleosides (16) to react selectively with aminoacyl-thiols (1).

Nucleoside aminoacylation was most rapid at pH 8.0–9.0 but was most 2′,3′-diol selective and highest yielding at pH 6.0–7.0. The aminoacyl nucleoside products (17) were also most stable at lower pH and so, overall, ribonucleoside aminoacylation was most effective at pH 6.5 (Fig. 2c). Aminoacylation was observed across a wide range of

nucleoside concentrations (2–100 mM) and aminoacyl-thiol concentrations (20–600 mM) at room temperature. As expected, a higher (steady-state) concentration of aminoacyl nucleoside (17) was observed when fuelled by a higher concentration of aminoacyl-thiol 1. However, notably, on freezing, the initial conditions could be extremely dilute (1–10 mM) and result in a very high yield of aminoacyl-RNA (17). For example, incubating a frozen solution of uridine 16U (2 mM) and thioester 1$_{Ala}^e$ (10 mM) at −7 °C led to alanyl-uridine 17$_{Ala}^U$ (58%) in excellent yield (Fig. 2d and Supplementary Table 12), despite the initial and final very low concentration of the solution. It is of note that eutectic phase conditions also promote ribozyme activity[27], warranting further investigation of eutectic phase aminoacyl-thiol reactions.

Aminoacylation was kinetically selective for the furanosyl-2′,3′-diol, probably because of its increased acidity with respect to other alcohols. To explore this selectivity further, we investigated the reaction of L-1$_{Ala}^e$ with a range of nucleosides (Fig. 2c and Extended Data Fig. 1). As expected, aminoacylation was nearly completely suppressed for 2′,3′-dideoxyadenosine 20A, which only has a primary 5′-alcohol and no diol. Canonical 2′-deoxynucleosides (DNA; 21) also furnished very low yields of aminoacylation. However, non-canonical 3′-deoxyadenosine 22A and 5′-deoxyuridine 23U both underwent substantial aminoacylation. This initially suggested that RNA may react through its 2′-alcohol, followed by equilibration of the 2′-ester and the 3′-ester. However, further investigation revealed that thioester 1 reacted effectively and selectively with 3′-O-methoxy-adenosine 24A and 2′-O-methoxy-adenosine 25A at their secondary alcohols, which indicated that the 2′-alcohol and 3′-alcohol of RNA are comparably reactive. Together, these results demonstrate a strong kinetic preference for reaction at the secondary alcohol of ribonucleosides, rather than at the sterically more accessible primary alcohol, in sharp contrast to the well-known selectivity observed in organic synthesis.

Next, we investigated the effect of nucleoside phosphorylation on aminoacylation. Nucleoside-5′-monophosphates (NMP; 26), nucleoside-5′-diphosphates (NDP; 27) and nucleotide-5′-triphosphates (NTP; 28) were all much less reactive than nucleosides (16), whereas overall levels of thioester turnover (to amino acid) remained similar. This indicated that nucleoside phosphate (NXP) aminoacylation was slower, rather than aminoacyl-ester hydrolysis being faster. We suspected that NXP aminoacylation was inhibited by the elevated diol p$K_a$ following phosphorylation. In line with this reasoning, negligible aminoacylation of adenosine-3′-phosphate (29A) and adenosine-2′-phosphate (30A) was observed; consistent with the high p$K_a$ of their 2′-hydroxyl and 3′-hydroxyl relative to a 2′,3′-diol. Notably, this phosphate-suppressed reactivity was found to be specific to phosphate monoesters and polyphosphates. Nucleoside phosphodiesters were observed to undergo comparable aminoacylation to unphosphorylated nucleosides. Therefore, adenosine-3′,5′-cyclic phosphate (31A) underwent highly effective 2′-aminoacylation, whereas 5′-aminoacylation of adenosine-2′,3′-cyclic phosphate 32A remained ineffective, further demonstrating the pronounced kinetic selectivity for reaction at the secondary over the primary alcohol in nucleotides (Extended Data Fig. 1).

The suppressed aminoacylation of 2′,3′-diols in NXPs (but not phosphodiesters) could play an important role in directing aminoacylation to oligonucleotide-2′,3′-diols, which are necessarily 5′-phosphodiesters, under conditions in which phosphorylated NXP monomers are concurrently available for RNA synthesis. The difference between phospho-monoester and phospho-diester aminoacylation would (correctly) direct aminoacylation within a network of reactions (that is, in a cell) that can simultaneously synthesize RNA and exploit RNA aminoacylation in peptide synthesis.

## Selective nucleic acid aminoacylation

Control over the site of aminoacylation in oligonucleotides is essential for optimal positioning of aminoacyl esters within RNA. Life specifically

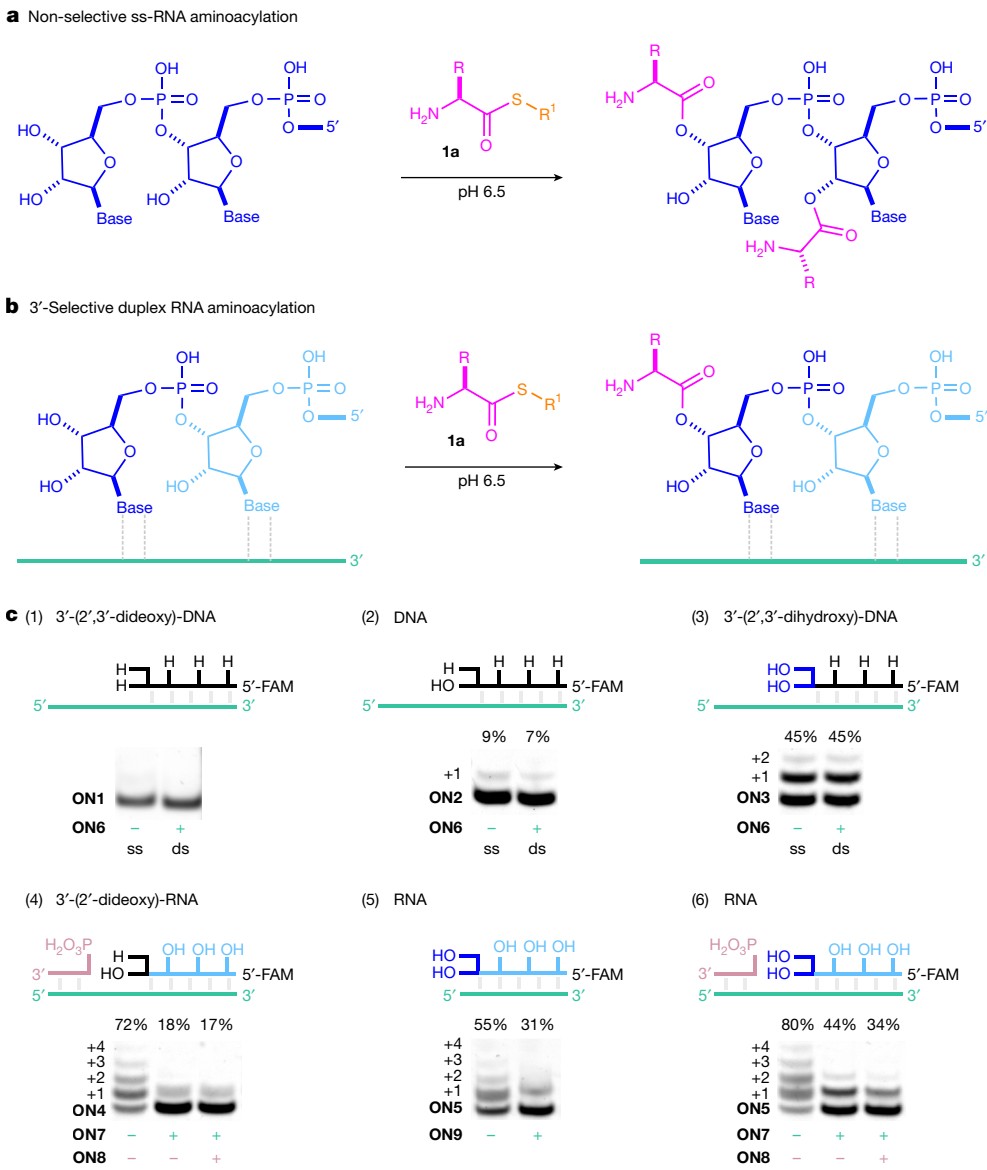

**Fig. 3 | Selective aminoacylation of oligonucleotides in water.**
**a**, Aminoacylation of single-stranded (ss) RNA at terminal 2′,3′-diol and internal 2′-alcohols. **b**, Selective 2′,3′-aminoacylation of duplex (ds) RNA by aminoacyl-thiols (**1a**, -SR¹ = pantetheinyl) in water. **c**, PAGE images showing the reaction of 5′-fluorescein (FAM) labelled ss-/ds-**ON1**–**ON5** (0.5 µM) with pantetheinyl-alanine L-$1^a_{Ala}$ (200 mM or 600 mM) in MES buffer (200 mM, pH 6.5) and KCl (500 mM) after 16 h at room temperature: (1) 5′-FAM-dUdAdGdGdAdGdAdGdCddC (**ON1**) with L-$1^c_{Ala}$ (200 mM), ±5′-dGdCdAdGdUdUdGdGdUdCdUdCdCdUdA (**ON6**);

(2) 5′-FAM-dUdAdGdGdGdAdGdAdCdCdA (**ON2**) with L-$1^a_{Ala}$ (600 mM), ±**ON6**;
(3) 5′-FAM-dUdAdGdGdAdGdAdCdCA (**ON3**) with L-$1^a_{Ala}$ (600 mM), ±**ON6**;
(4) 5′-FAM-UAGGAGACCdA (**ON4**) with L-$1^a_{Ala}$ (600 mM), ±5′-GCAGUUGGUCUCCUA (**ON7**) and ±5′-pACUGC (**ON8**); (5) 5′-FAM-UAGGAGACCA (**ON5**) with L-$1^a_{Ala}$ (200 mM), ±5′-UGGUCUCCUA (**ON9**); (6) **ON5** with L-$1^a_{Ala}$ (600 mM), ±**ON7** and ±**ON8**. See Supplementary Figs. 102–105 and Supplementary Tables 16–18 for PAGE images showing ss-/ds-**ON2**–**ON5** with L-$1^a_{Ala}$ (200 mM).

aminoacylates RNA at the 3′-terminus in the first step of protein synthesis (Fig. 1a) and therefore we next turned our attention to nucleic acid aminoacylation, for which internal 2′-aminoacylation introduced a further challenge to selectivity[28].

We first investigated the aminoacylation of DNA oligonucleotides (**ON1**–**ON3**). Minimal DNA aminoacylation was observed, as a single-gel band, consistent with a single site of aminoacylation at the DNA 3′-terminus (**ON2**; Fig. 3c (2)). As expected, aminoacylation was blocked by a terminal 2′,3′-dideoxynucleotide (**ON1**; Fig. 3c (1)) but strongly (sixfold) enhanced by a terminal 2′,3′-diol (**ON3**; Fig. 3c (3)). As well as enhanced yield with the 2′,3′-diol terminus, a second minor bis-aminoacylation product was observed, demonstrating the greatly enhanced reactivity of the (RNA) 2′,3′-diol over the (DNA) 3′-hydroxyl in oligonucleotides.

We next investigated (single-strand) ss-RNA (**ON4** and **ON5**). Extensive and non-specific ss-RNA aminoacylation was observed, with aminoacylation at several hydroxyls throughout the RNA oligomer (**ON5**; Fig. 3c (6)). Multiple-site ss-RNA aminoacylation was not blocked by a 3′-terminal deoxynucleotide (**ON4**; Fig. 3c (4)) and little or no selectivity was observed for aminoacylation at the 2′,3′-diol of ss-RNAs, even in a short nucleic acid. Longer RNAs have more internal 2′-hydroxyls and further diminish the selectivity of aminoacylation. The internal aminoacylation prevents the selective aminoacylation of ss-RNA at the 3′-terminus that is required for RPS.

Concerned by the poor selectivity of ss-RNA, and considering that nucleic acid replication must furnish nucleic acid duplex, we next investigated (double-stranded) ds-RNA. We were pleased to find that the formation of Watson–Crick duplex resurrected selective

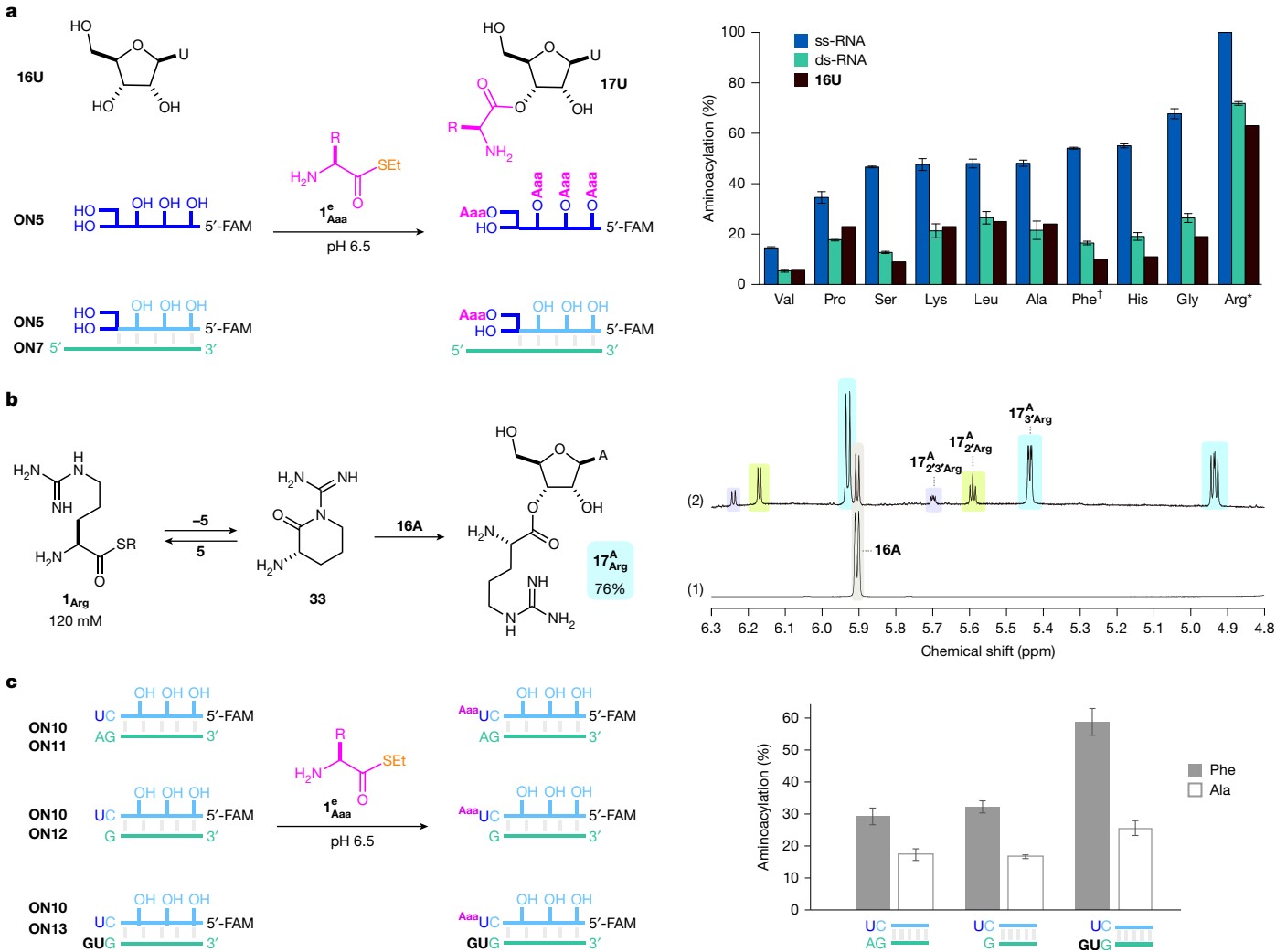

**Fig. 4 | Aminoacyl-thiol side-chain scope. a**, Aminoacylation (%, mean ± s.d.; $n = 3$) for the reaction of: black, **16U** (20 mM) in water at pH 6.5 and room temperature after 24 h; blue, ss-**ON5** (0.5 μM) in MES buffer (400 mM, pH 6.5) and KCl (1 M) with L-$1_{Aaa}^e$ (200 mM) at room temperature after 16 h; and green, ds-**ON5/ON7** (0.5 μM) in MES buffer (400 mM, pH 6.5) and KCl (1 M) with L-$1_{Aaa}^e$ (200 mM) at room temperature after 16 h. †L-$1_{Phe}^e$ (100 mM). *L-$1_{Arg}^e$ (60 mM). See Fig. 5c for L-$1_{Asp}^e$ and Supplementary Figs. 81–83 for L-$1_{Glu}^e$, L-$1_{Gln}^e$ and L-$1_{Met}^e$. **b**, The reaction of arginine thioester L-$1_{Arg}^e$ was enhanced by side-chain catalysis. $^1$H NMR spectra show: (1) adenosine **16A** (2 mM); (2) the reaction of **16A** (2 mM) with L-$1_{Arg}^e$ (120 mM) at pH 6.5 and room temperature after 6 h, which formed

arginyl-adenosine $17_{Arg}^A$ (76%) in excellent yield. See Fig. 1b (2) for L-$1_{Arg}^e$ reaction time course. The Arg side chain (p$K_{aH}$ 13.8) is drawn neutral but will be fully protonated at pH 6.0–7.0. **c**, Minizyme-enhanced aminoacylation of oligonucleotide 5′-FAM-UGAGAGAGCCU (**ON10**). Aminoacylation (%, mean ± s.d.; $n = 3$) of **ON10** as a blunt end duplex (+5′-AGGCUCUCUCA (**ON11**) 0.5 μM), underhang (+5′-GGCUCUCUCA (**ON12**) 0.5 μM) and 'minizyme' complex (+5′-GUGGCUCUCUCA (**ON13**) 0.5 μM) with L-$1_{Phe}^e$ or L-$1_{Ala}^e$ (100 mM) in MES buffer (200 mM, pH 6.5) and KCl (500 mM) after 16 h at room temperature. See Supplementary Figs. 122–124 for further experimental details.

aminoacylation. The yield of 2′,3′-diol aminoacylation was equal in ds-RNA (**ON5**) and ss-oligomer **ON3**, a chimeric oligonucleotide with a single RNA at the 3′-terminus (Fig. 3c), providing a clear demonstration that duplex formation did not inhibit aminoacylation of the RNA 2′,3′-diol. However, at the same time, duplex formation strongly inhibits aminoacylation of internal 2′-hydroxyls[29] (Fig. 3c (6)) and therefore aminoacylation is highly selective for the terminal 2′,3′-diol in ds-RNA. Surprised by this simple and inherent mechanism by which RNA duplex directs selective aminoacylation, we next investigated a more congested nicked duplex, with an adjacent downstream 5′-phosphorylated oligomer. The presence of this downstream oligomer did not block diol aminoacylation (Fig. 3c), demonstrating that selective 2′,3′-aminoacylation can occur despite a proximal 5′-phosphate. These results indicate that RNA's ability to form duplexes may be essential to directing the site of RNA aminoacylation, as well as ordering the alignment of tRNAs on mRNA templates during protein synthesis.

## Side-chain compatibility of RNA aminoacylation

Given the marked efficiency of 3′-selective ds-RNA aminoacylation, we next investigated whether changing the aminoacyl-thiol side chain would enable a broad scope of chemical aminoacylation of RNAs in water with proteinogenic amino acids. Crucially, thioester-mediated ribonucleoside aminoacylation was extremely efficient relative to the poor reactivity shown with amines (Fig. 2). This led us to consider that side-chain compatibility, which has not been previously examined in other aminoacylation studies, would be high.

Aminoacyl-thiols with simple or lipophilic side chains ($1_{Gly}^e$, L-$1_{Leu}^e$, L-$1_{Pro}^e$) reacted similarly to alanine thioester $1_{Ala}^e$, although, as expected, β-branched thioester $1_{Val}^e$ reacted sluggishly (Fig. 4a). Thioesters with weakly nucleophilic side chains (L-$1_{His}^e$, L-$1_{Glu}^e$, L-$1_{Gln}^e$, L-$1_{Met}^e$, L-$1_{Ser}^e$) formed aminoacyl nucleosides **17** in good, albeit slightly lower, steady-state yield. Notably, no serine acylation was observed despite a tenfold excess

of the (Ser) α,β-amino-alcohol over nucleoside-diol, further demonstrating high selectivity for the 2′,3′-diol.

Notably, arginine thioester L-$1^e_{Arg}$ was an extremely effective aminoacylating agent, furnishing arginyl-adenosine $17^A_{Arg}$ in up to 76% yield (Fig. 4b). Quantitative ss-RNA and 64% 3′-selective ds-RNA arginylation was observed following reaction of L-$1^e_{Arg}$ with oligomeric RNAs. Further analysis indicated that cyclic arginine **33** was formed in situ and **33** aminoacylated RNA even more effectively than thioester **1** (Supplementary Table 13), demonstrating pronounced intramolecular nucleophilic catalysis. To our knowledge, this mode of nucleophilic Arg catalysis has not previously been reported in enzyme catalysis but it is highly likely that acyl-transfer enzymes would have exploited this catalytic strategy. The enhanced aminoacylation of RNA with arginine, as well as the side-chain-specific affinity of Arg-peptides for RNAs[30–32], warrant further investigation of the structure and function of arginylated RNAs.

Given this unexpected cyclization with Arg, a side chain normally considered non-nucleophilic, we might expect lysine thioester L-$1^e_{Lys}$ to cyclize rather than aminoacylate. However, aminoacylation with L-$1^e_{Lys}$ proceeded just as effectively as aminoacylation with alanine thioester L-$1^e_{Arg}$. Overall, excellent proteinogenic side-chain tolerance was observed during nucleoside aminoacylation, suggesting that aminoacyl-thiol (**1**) is a privileged substrate for the aminoacylation of RNA in water.

Comparable aminoacylation yields were observed for ds-RNAs terminating in all four canonical ribonucleotides (A, U, C and G; **ON5, ON14−ON17**) with the same thioester (Supplementary Fig. 112), indicating the propensity of thioester-mediated aminoacylation to charge a variety of proteinogenic amino acids onto ds-RNA in a sequence-independent manner. However, notably, 2′,3′-aminoacylation was found to be more effective (about 1.3−1.5-fold) with a template overhang at the 3′-terminus, relative to a nicked duplex, suggesting a potential catalytic role for the template strand. Indeed, we found that ribozymes augmented thioester-mediated side-chain-selective aminoacylation. Specifically, an RNA duplex (**ON10** + **ON13**) with a mispaired 3′-U/5′-U and 5′-G overhang[15], amplified 3′-aminoacylation yields (Fig. 4c) with respect to ds-RNA (**ON10** + **ON11**) and enabled substantial aminoacylation with phenylalanine thioester (L-$1^e_{Phe}$), even at extremely low (2 mM) concentrations (Supplementary Fig. 124 and Supplementary Table 26). These results indicate that enhancing the 2′,3′-diol-selective reactivity of specific amino thioesters (**1**) with specific RNA catalysts has the potential to direct amino acid sequence pairings, paving the way for a (primitive) coding system.

Ideally, RNA aminoacylation would be intrinsically directed to the (controlled) products of RNA biosynthesis (that is, RNAs with 2′,3′-diols) and adventitious aminoacylation of RNA degradation products would be inhibited. So we next considered the effect that RNA degradation would have on aminoacylation. RNA hydrolysis specifically furnishes RNAs that terminate in 2′-phosphate or 3′-phosphate, rather than the 5′-phosphates that are produced by RNA biosynthesis. Therefore we next incubated **ON5-3′p** with thioester **1** and were pleased to observe that the 3′-phosphate very strongly suppressed ds-oligonucleotide aminoacylation (Extended Data Fig. 2). Notably, this inhibition offers an innate mechanism for RNA hydrolysis to prevent adventitious aminoacylation, which would be highly beneficial within a system (or cell) that is continuously generating and degrading (catalytic) RNAs.

In a preliminary investigation of the diastereoselectivity of thioester-mediated 2′,3′-aminoacylation, oligonucleotides terminating in pyrimidines were observed to react with negligible selectivity (**ON15**, **ON16**: <0.06 d.e.), whereas those ending with a purine led to a slightly increased d.e. (**ON14**, **ON17**: about 0.1 d.e.). This selectivity was not affected by duplex formation and FAM-(dN)₉N chimeric nucleic acids (for example, **ON2**) underwent aminoacylation with similar diastereoselectivity (<0.2 d.e.). This selectivity seems to be governed at

the monomeric level, with comparable selectivity observed during the aminoacylation of nucleosides; pyrimidines **16C/16U** = <0.05 d.e., purines **16A/16G** = 0.3 d.e. with $1^a_{Ala}$ (Supplementary Table 25). Given the transient nature of aminoacyl esters, as only one intermediate of nucleic-acid-mediated peptide synthesis, the impact of this selectivity on the overall multistep scheme cannot yet be known, but it is inevitable that (several) subsequent steps will affect diastereoselectivity[33] and even evolved synthetase enzymes may not achieve strictly L-selective aminoacylation[34,35], despite the presence of D-amino acids in the cell. Encoding amino acids during aminoacylation will inevitably affect stereochemical selection but we believe that stereochemically agnostic tRNAs are likely to be important to enable the comprehensive side-chain compatibility necessary for uniformly effective synthesis of diverse peptides, with sterically diverse amino acid side chains, which are encoded only by nucleic acid information (rather than being intrinsically favoured by peptide structures) during RPS. Therefore, we have not attempted to augment or reverse the innate (lack of) diastereoselectivity observed here but catalytic aminoacylation would pave the way to address amino acid coding and selection in the future, for which amino acid chirality can be set during the synthesis of the aminoacyl-thiol and so we next turned our attention to thiol-catalysed aminoacyl transfer.

## Thiol-catalysed aminoacylation

Given the unprecedented selectivity for nucleoside aminoacylation by aminoacyl-thiols (**1**), we next investigated their potential provenance, as although aminoacyl-thiols (**1**) are biological, no reasonable non-enzymatic route had yet been found to form these fascinating compounds. We suspected that activated carboxylates, such as the biological aminoacyl adenylate **2** or prebiotic NCAs **4**, would react with thiols (**5**) to form thioester **1** in situ, which could then aminoacylate nucleic acids with a selectivity, efficiency and side-chain scope that would not be possible without a thiol cofactor.

To test this hypothesis, we incubated a range of prebiotically plausible NCAs **4** with thiols **5**. We observed rapid and highly efficient synthesis of aminoacyl-thiol **1**, even under highly dilute (1 mM) conditions (Fig. 5a and Extended Data Fig. 3a). Simple and hydrophobic NCAs (**4**$_{Gly}$, L-**4**$_{Phe}$, L-**4**$_{Val}$) furnished their respective aminoacyl-thiol (**1**) in near-quantitative yield on reaction with coenzyme M (thiol **5c**). Biological aminoacyl-phosphate **2**$_{Ala}$ (20 mM) also reacted with 3-mercaptopropionic acid (thiol **5b**) to furnish aminoacyl-thiol$1^b_{Ala}$ in good unoptimized yield (42%; Extended Data Fig. 3b and Supplementary Fig. 166). Notably, without thiols, NCAs (**4**) underwent self-condensation and hydrolysis[36], which completely blocked nucleoside aminoacylation, but spontaneous self-condensation of activated amino acids (**2**, **4**) was strongly suppressed by thiol **5**.

Thiol-suppressed NCA self-condensation suggested that thiol-catalysed aminoacylation of RNA-2′,3′-diols might be feasible, so next we incubated Ala-NCA (L-**4**$_{Ala}$) with thiol **5c** and adenosine (**16A**) in water. After 16 h, the resultant solution was pH 6.0 and we observed ribonucleoside aminoacylation furnishing alanyl-adenosine $17^A_{Ala}$ in 35% yield (Fig. 5b (2)). This effective thiol-mediated aminoacylation stands in sharp contrast to the reaction in the absence of the thiol cofactor; with no thiol, no nucleoside aminoacylation was observed under otherwise identical conditions (Fig. 5b (1)). As expected, on incubation of oligonucleotides with NCAs (**4**) no oligonucleotide aminoacylation was observed. However, the addition of thiol **5b**, alongside NCA **4**$_{Ala}$, led to effective oligonucleotide aminoacylation, through in situ formation of aminoacyl-thiol **1**. For example, oligonucleotide (**ON5**) reacted with L-**4**$_{Ala}$ (400 mM) and thiol **5b** (200 mM) in MES buffer (pH 6.0) to yield aminoacyl-RNA (23%). No aminoacylation of **ON5** was observed in the absence of thiol **5b** (Supplementary Fig. 127), demonstrating the marked enhancement of aminoacylation by thiols (**5**).

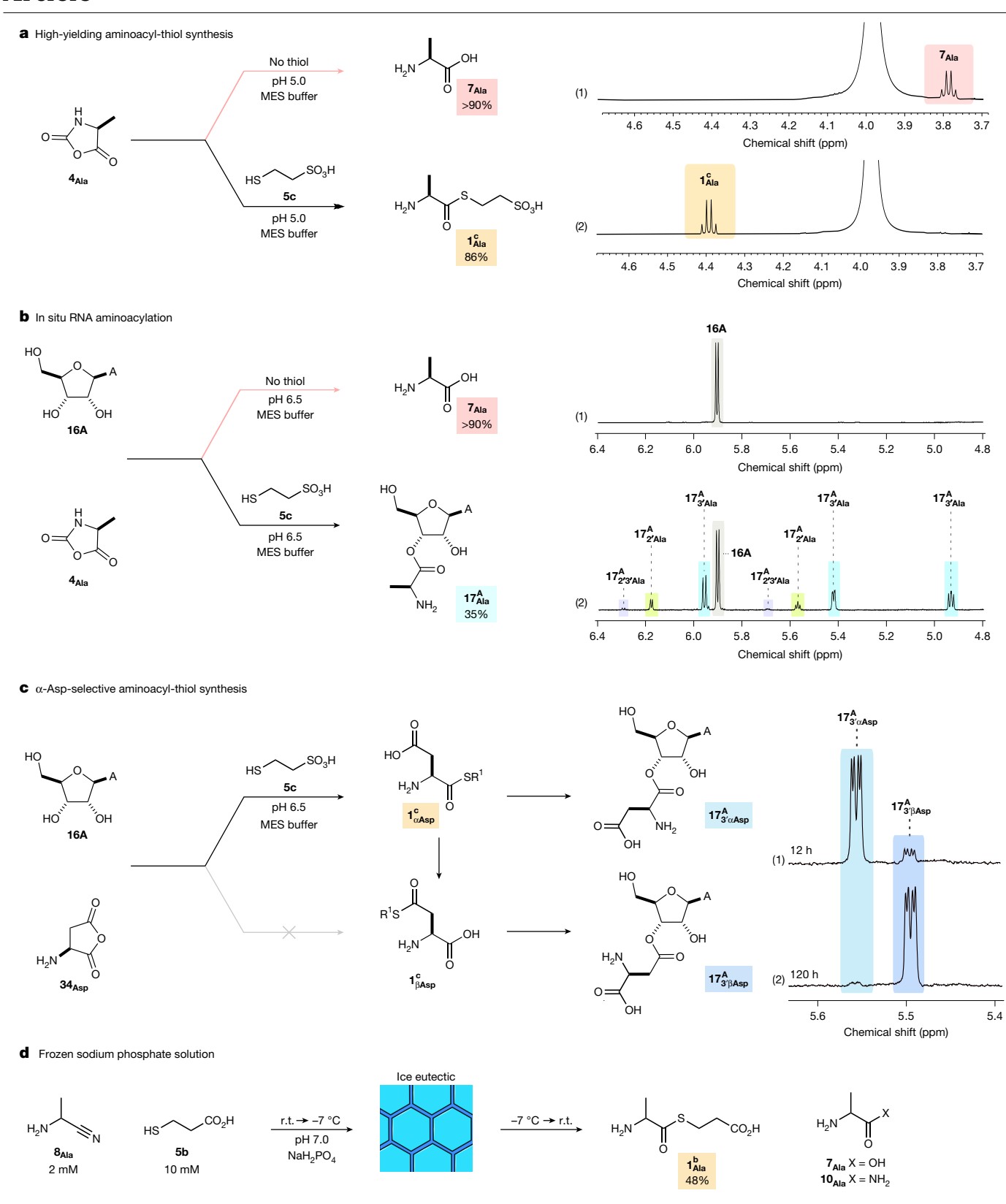

**a** High-yielding aminoacyl-thiol synthesis

**b** In situ RNA aminoacylation

**c** α-Asp-selective aminoacyl-thiol synthesis

**d** Frozen sodium phosphate solution

**Fig. 5 | Thioester-mediated synthesis of aminoacyl nucleosides.** ¹H NMR spectra showing various reactions. **a**, NCA L-**4**_Ala (20 mM) in MES buffer (200 mM; 3.99 ppm), at pH 5.0 and room temperature, which yields: (1) alanine L-**1**$^c_{Ala}$ (>90%) without (coenzyme M) thiol **5c**, and (2) alanine thioester L-**1**$^c_{Ala}$ (86%) with thiol **5c** (100 mM), the aminoacyl-thiol product is highly stable at pH 5.0. **b**, Adenosine **16A** (20 mM) and NCA L-**4**_Ala (600 mM) at pH 6.5 and room temperature, for which no nucleoside aminoacylation was observed. Adenosine **16A** (20 mM), NCA L-**4**_Ala (600 mM) and thiol **5c** (1 M) at pH 6.5 and room temperature, which

furnished alanyl-adenosine **17**$^A_{Ala}$ (35%) from the reaction of adenosine (**16A**) with aminoacyl-thiol (L-**1**$^c_{Ala}$) formed in situ in water. R¹ = CH₂CH₂SO₃H. **c**, Adenosine **16A** (20 mM), aspartate-anhydride **34**_Asp (600 mM) and thiol **3c** (1 M) at pH 6.5 and room temperature after (1) 0.5 days and (2) 5 days, which furnished aspartyl-adenosine **17**$^A_{Asp}$ from the reaction of adenosine (**16A**) with aminoacyl-thiol L-**1**$^c_{Asp}$ formed in situ in water. **d**, Incubation of prebiotically plausible α-aminonitrile **8**_Ala (2 mM) and thiol **5b** (10 mM) in frozen sodium phosphate solution (pH 7.0, 50 mM) at −7 °C for 30 days yields aminoacyl-thiol **1**$^b_{Ala}$ in 48% yield at neutral pH.

We next turned our attention to glutamate (Glu) and aspartate (Asp), which have two carboxylate moieties. Glutamate thioester L-$1^e_{Glu}$ (and Glu-NCA $4_{Glu}$ + thiol $5c$) were observed to only yield α-aminoacylation of nucleosides (11% $17^A_{Glu}$ from $4_{Glu}$ through in situ formation L-$1^c_{Glu}$; Supplementary Fig. 160). Glu-isomerization rapidly quenched γ-activation through pyroglutamate formation, which completely blocked γ-aminoacylation. Aspartate has two adjacent carboxylates that cannot lactamize and therefore Asp is the most challenging proteinogenic amino acid disposition for chemoselective activation and α-selective aminoacylation. However, Asp-NCA ($4_{Asp}$) reacted with coenzyme M (thiol $5c$) to yield aminoacyl-thiol L-$1^c_{αAsp}$ (49%) with excellent α-selectivity (α/β 11:1) and in situ reaction with adenosine ($16A$) yielded aspartyl-adenosine $17_{αAsp}$ also with excellent α-selectivity after 18 h (Supplementary Fig. 162). Aspartyl-thioester L-$1^c_{αAsp}$ slowly isomerized to its β-isomer L-$1^c_{βAsp}$, which then led to selective β-aminoacylation after 5 days. However, initial α-selectivity was strongly favoured. Moreover, the α-selectivity of Asp was not only favoured by the inherent (α-amine-tethered) α-carboxylate activation of NCA $4$ but also for Asp-anhydride $34_{Asp}$, despite dual activation of both the α-carboxylate and β-carboxylate. Notably, Asp-anhydride $34_{Asp}$ reacted with thiols ($5$) to selectively yield α-aspartyl-thioester L-$1^c_{αAsp}$ (45%) with excellent α-selectivity (α/β 12:1) in water after 30 min (Supplementary Fig. 165). Therefore, incubating Asp-anhydride $34_{Asp}$, thiol $5c$ and adenosine $16A$ in water at pH 6.5 resulted in selective α-aminoacylation of $16A$ and furnished $17^A_{αAsp}$ (12%) after 12 h (Fig. 5c (1)), demonstrating a profound kinetic selectivity for natural (proteinogenic) α-aminoacylation, irrespective of α-amine-tethered activation.

The reaction of thiols $5$ with activated amino acids ($2$, $4$), which have previously been suggested to be relevant for prebiotic aminoacylation[15,16], illustrates the role that thiol cofactors could have played in enabling catalytically controlled aminoacylation. These results demonstrate that there are several high-yielding prebiotic pathways to aminoacyl-thiols $1$, as well as for in situ thioester-mediated ribonucleoside-2′,3′-aminoacylation at near-neutral pH. Aminoacyl-thiols $1$ seem to be very well suited to the spontaneous and selective aminoacylation of nucleosides in water.

Both aminoacyl adenylate $2$ and NCAs $4$ are derived from amino acids by means of electrophilic activation and neither has a long lifetime in water. Although this makes their efficient reaction with thiols more noteworthy, we envisaged that a prebiotically plausible route to aminoacyl-thiols ($1$) might be feasible by the reaction of aminonitriles ($8$) with thiols ($5$). Aminonitriles ($8$) are considered probable precursors to amino acids on the early Earth but their high-energy nitrile has sufficient energy to be directly transformed into a thioester, avoiding fully hydrolysed amino acids and the resultant need for an external source of chemical energy. Aminonitriles ($8$) are kinetically stable but we have found that thiols ($5$) can unlock the energy in nitriles to enable formation of peptide bonds[37–39]. Furthermore, we have recently discovered a selective route by which pantetheine ($5a$), the universal biological thioester-bearing moiety that is central to non-ribosomal peptide synthesis, forms in water from nitriles[20]. This link between nitriles, biological peptides and biological thiols led us to consider the direct link between nitriles and activated amino acid thioesters more closely.

We began by studying the reaction of alanine nitrile ($8_{Ala}$) with 3-mercaptopropionic acid ($5b$)[33,38]. No thioester was observed at pH 7.0–9.0 (Supplementary Discussion and Supplementary Fig. 185) but under mildly acidic conditions (pH 5.0), alanine thioester ($1^b_{Ala}$) was formed in 20% yield (Extended Data Fig. 4a). A range of simple and biological thiols, including pantetheine ($5a$) and coenzyme M ($5c$), reacted with aminonitriles ($8$) to yield aminoacyl-thiols ($1$) under acidic conditions (Extended Data Fig. 5a).

The formation of thioester $1$ from aminonitrile $8$ was observed at prebiotically plausible pH (that is, pH 5.0), but it was most effective under slightly more acidic conditions. At pH 3.0–4.0, the formation of aminoacyl-thiol $1$ was observed in 42–48% yield. However, initially, this low pH (<5.0) seemed prebiotically unrealistic. We reasoned that the selectivity for aminoacyl-thiol $1$ would be improved by general-acid catalysis[40] and phosphate was found to modestly increase the yield of thioester $1^b_{Ala}$ (56%; Extended Data Fig. 5). However, phosphate, as well as being a catalyst for thioester formation, could drive (reversible) pH change from neutral pH to mildly acidic conditions by a highly plausible prebiotic mechanism, because freezing phosphate solutions leads to an ideal drop in pH (ref. 41), to the sweet spot for aminoacyl-thiol $1$ formation. Accordingly, we found that solutions of nitrile $8_{Ala}$ and thiol $5b$, buffered by phosphate at pH 7.0, yielded alanine thioester $1^b_{Ala}$ efficiently when frozen and incubated at −7 °C. Furthermore, freezing allows the initial conditions to be, in principle, almost indefinitely dilute owing to the concentrating effects of forming the eutectic phase. For example, we observed that incubating 2 mM alanine nitrile $8_{Ala}$ with 10 mM thiol $5b$ at −7 °C led to effective synthesis of alanine thioester $1^b_{Ala}$ (48%) after 30 days and a neutral pH solution of thioester $1$ on thawing the frozen solution (Fig. 5d and Supplementary Fig. 195), demonstrating low concentration, prebiotically plausible, synthesis of thioester $1$ from a neutral pH solution of aminonitrile $8$.

A wide range of aminonitriles ($8_{Arg}$, $8_{Gly}$, $8_{Leu}$, $8_{Lys}$, $8_{Met}$, $8_{Pro}$ and $8_{Ser}$) underwent good conversion to their respective thioesters $1^b_{Aaa}$ (40–64%; Extended Data Fig. 5, entries 11–19). However, the conversion of $8_{Phe}$ (21%) and $8_{Val}$ (14%) were lower, probably because of thioimidate tautomerization (Supplementary Discussion). Proteinogenic α-aminonitriles ($8$) were readily converted in comparable yield to their respective aminoacyl-thiols ($1$). Notably, however, non-proteinogenic α-hydroxy nitrile $36$, β-aminonitrile $37$ and acylated α-aminonitrile $38$ all underwent <5% reaction with thiol $5b$ under the same conditions (Extended Data Fig. 4b–d), demonstrating notable selectivity for the formation of aminoacyl-thiols of proteinogenic amino acids.

## Selective peptidyl-RNA synthesis

Nucleoside aminoacylation with aminoacyl-thiols ($1$) results from judiciously balanced reactivity of the nucleoside-2′,3′-diol and aminoacyl-thiol $1$, allowing (biologically) activated amino acids (that is, thioester $1$) to chemoselectively deliver aminoacyl-RNA in water (Fig. 1). Notably, thioester activation is orthogonal to peptide bond formation at neutral pH. To demonstrate further the importance of thioester activation for this selectivity, we set out to undertake an investigation of peptide synthesis and the formation of peptidyl-RNA ($40$) in water under the same conditions that yield aminoacyl-RNA ($17$).

First we tested the direct synthesis of peptidyl-RNA ($40$) by means of the reaction of thioester-activated peptides with RNA. Notably, reaction at the RNA-2′,3′-diol was blocked by N-acylation of thioester $1$ and no peptidyl-RNA ($40$) was observed on incubating nucleoside $16A$ with N-acetyl-aminoacyl-thiol Ac-Ala-SEt $39^e_{Ala}$ at pH 6.5 (Extended Data Fig. 6 (i)). Moreover, incubating $16A$ with L-$1^e_{Ala}$ and $39^e_{Ala}$ led to the selective formation of aminoacyl-adenosine ($17^A_{Ala}$), whereas again no formation of peptidyl-RNA was observed (Extended Data Fig. 6 (ii)). This further demonstrates the importance of the amine moiety for promoting selective reaction at the RNA-2′,3′-diol but, more importantly, selective reaction of the free amine (that is, thioester $1$) provides a mechanism to promote peptide formation by means of the nucleophilicity of the aminoacyl-RNA ($17$) as required for nucleic-acid-orchestrated peptide synthesis (that is, RPS).

Aminoacylated nucleotides (for example, $17$) have been demonstrated to form peptides by iterative, non-enzymatic cycles of 1-ethyl-3-(3-dimethylaminopropyl)carbodiimide (EDC)-mediated condensation and so the selective formation of aminoacyl-RNAs ($17$) from aminoacyl-thiols ($1$) completes the non-enzymatic, protecting-group-free RNA-mediated synthesis of peptides from

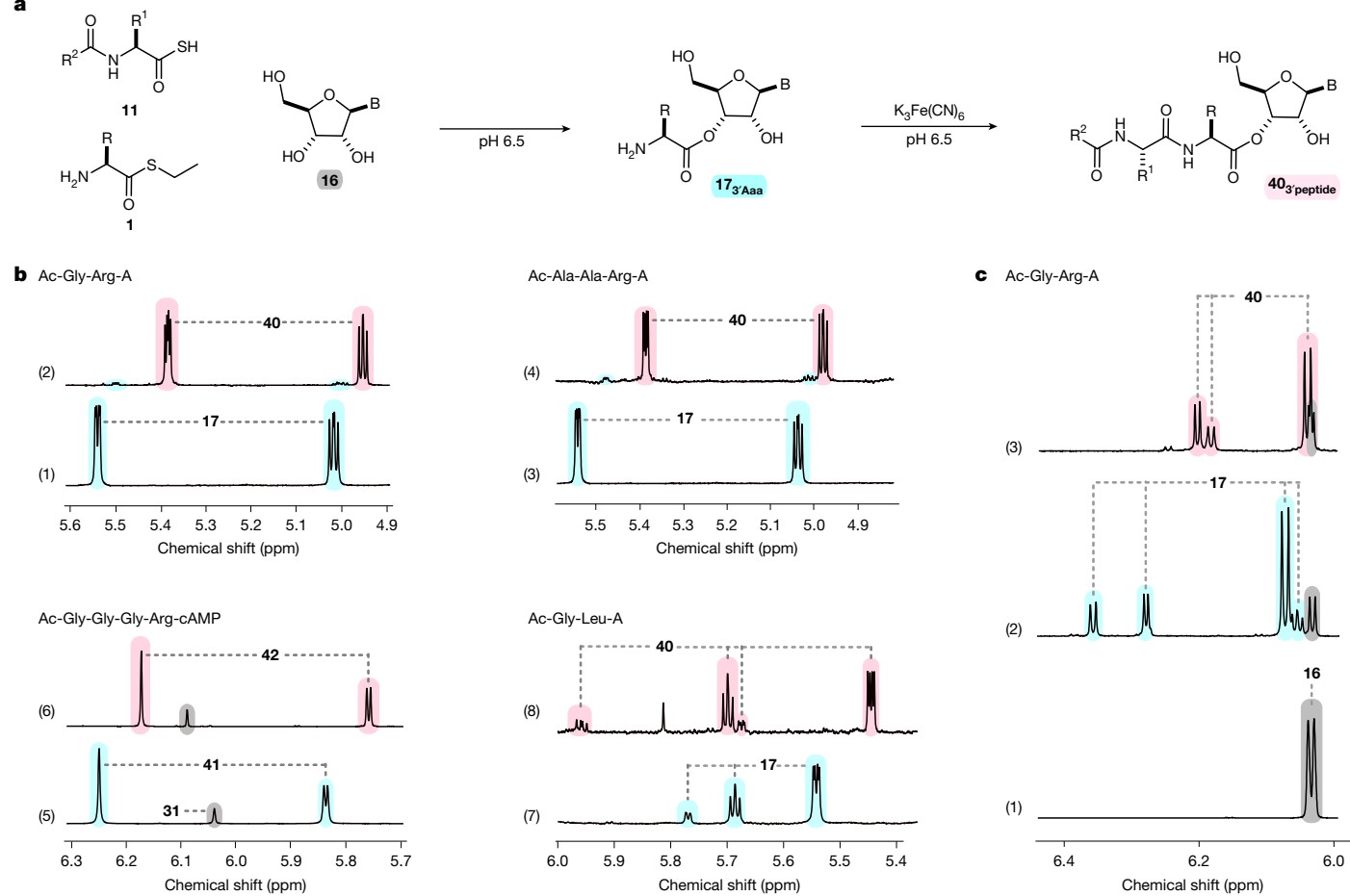

**Fig. 6 | One-pot synthesis of peptidyl-RNA in water. a**, Incubation of **16** with thioester **1** and thioacid **11** yields aminoacyl-RNA **17**, and in situ oxidation of **11** converts **17** into peptidyl-RNA **40** in near-quantitative yield. R and $R^1$ = peptide side chain. $R^2$ = Me or peptide. B = nucleobase. **b**, $^1$H NMR spectra of: (1) **16A** (20 mM) and **33** (80 mM) in MES buffer (500 mM, pH 6.5) after 2 h; (2) after addition of $K_3Fe(CN)_6$ (150 mM) and thioacid **11**$_{Gly}$ (50 mM), which yields **40**$^A_{ArgGlyAc}$ (93%); (3) **16A** (20 mM) and **33** (80 mM) in MES buffer (500 mM, pH 6.0) after 3 h; (4) after addition of $K_3Fe(CN)_6$ (450 mM) and Ac-Ala-Ala-SH (**11**$_{AlaAla}$, 150 mM), which furnished **40**$^A_{ArgAlaAlaAc}$ (89%); (5) adenosine-3′,5′-cyclic phosphate **31A** (20 mM) and **33** (80 mM) in MES buffer (500 mM, pH 6.5) after 1 h, which yields aminoacyl-RNA **41**; (6) after addition of $K_3Fe(CN)_6$ (300 mM)

and Ac-Gly-Gly-Gly-SH (**11**$_{GlyGlyGly}$, 100 mM), which furnished **42**$^A_{ArgGlyGlyGlyAc}$ (95%); see Extended Data Figs. 1 and 8 for structures of **31**, **41** and **42**; (7) **16A** (20 mM) and L-**1**$^e_{Leu}$ (200 mM) in MES buffer (1 M, pH 6.5) after 24 h; (8) after addition of $K_3Fe(CN)_6$ (1.05 M) and **11**$_{Gly}$ (350 mM), which furnished peptidyl-RNA **40**$^A_{LeuGlyAc}$ (95%). 5.81 ppm (s, 1H) resonance = glycyl-thioacid by-product (see ref. 24). **c**, $^1$H NMR spectra to show: (1) **16A** (20 mM); (2) **16A** (20 mM), **33** (80 mM) and **11**$_{Gly}$ (100 mM) in MES buffer (1 M, pD 6.5) after 2 h; (3) after addition of $K_3Fe(CN)_6$ (300 mM), which yields **40**$^A_{ArgGlyAc}$ (96%). For a comparable one-pot peptidyl-RNA synthesis, in which **16A**, **33** and $K_3Fe(CN)_6$ are incubated to yield **17** and then thioacid **11**$_{Gly}$ was added at 2 h to furnish **40**, see Supplementary Fig. 250.

thioesters and RNA in water[42,43]. However, we considered it to be more important that, contrary to these EDC-activated results, in our reactions, aminoacyl-RNAs (**17**) are not observed to yield peptidyl-RNA (**40**) at all. Despite the aminoacyl-RNA (**17**) having a free amine and there being a large excess of thioester available, which could, in principle, react to yield peptides, no peptidyl-RNA (**40**) synthesis was observed, only the selective aminoacylation of RNA. This difference suggested that there must be an underlying chemical switch that would turn on peptide synthesis and this might be exploited to chemically control stepwise biomimetic synthesis of peptidyl-RNA in water.

We suspected that the difference resulted from the mode of activation. EDC-activated carboxylic acids are highly activated and so undergo reaction with amines to yield peptides in water. This stands in sharp contrast to the reactivity of aminoacyl-thiols (**1**) that are weakly activated and do not yield peptides under the same conditions. On reflection, we reasoned that peptide synthesis would be most effectively promoted by a change in the characteristics of acyl-activation, rather than by nucleic acid templating, and to test this hypothesis, we

next studied the synthesis of peptidyl-RNA from (prebiotically plausible) thioacids (**11**).

Thioacids are readily accessible from thioesters, on reaction with hydrogen sulfide ($H_2S$) at neutral pH (Fig. 2a), and can be activated selectively in water by ferricyanide, copper salts or cyanoacetylene ($HC_3N$) to undergo chemoselective, high-yielding peptide ligation. These ligations are extremely selective for peptide coupling and tolerate all of the 20 proteinogenic amino acid residues[24], therefore we suspected that they would be ideally suited to peptidyl-RNA synthesis in water.

Notably, aminoacyl-RNA (**17**) reacted with peptide thioacids **11** and ferricyanide at pH 6.0–6.5 to furnish peptidyl-RNA **40** in excellent to near-quantitative yield (Fig. 6). Unlike thioester **1**, activated thioacid **11** reacted very selectively with the amine moiety of **17**. Notably, unlike EDC activation, thioacid activation is orthogonal to peptide side chains and nucleoside functionalization[24] and so peptidyl-RNA **40** was observed to form very efficiently, without (thioacid-mediated) acylation of nucleoside alcohols or nucleobases (Extended Data Fig. 8). Moreover, owing to the mild (neutral) conditions, the stereochemistry of aminoacyl-thiols (**1**), aminoacyl nucleosides (**17**) and

peptide thioacids (**11**) were preserved during both aminoacylation and peptidyl-RNA synthesis in water; racemization was not observed and so peptidyl-RNAs (**40**) were formed as a single isomer from homochiral substrates.

Finally, owing to the orthogonal nature of thioester and thioacid activation and their capability to direct the two steps of RPS, we found that the one-pot synthesis of aminoacyl-RNA (**17**) in the presence of thioacid **11** was observed, at pH 6.5. No hydrolysis of **11** or peptide bond formation was observed (Fig. 6c (2)) and so subsequent in situ, one-pot oxidation of **11** then converted aminoacyl-RNA (**17**) into peptidyl-RNA (**40**) in near-quantitative yield (Fig. 6c (3) and Extended Data Fig. 7).

The selective formation of peptidyl-RNA **40** in water demonstrates the subtle, but important, difference in reactivity between aminoacyl-thiols **1** and peptide thioacids **11**, and shows conclusively that the selective reaction of thioester (**1**) with nucleoside alcohols cannot be explained by amine protonation at neutral pH. Thioesters and thioacids together provide the chemical differentiation required to control RNA aminoacylation and peptide synthesis, through orthogonal reactions, under the same (pH and concentration) conditions.

## Conclusion

The highly selective non-enzymatic synthesis of aminoacyl-thiols (**1**), coupled with our recent selective synthesis of pantetheine (**5a**)[20], completes a direct synthetic chain from simple nitriles to the aminoacyl-thiols used by life to facilitate non-ribosomal peptide synthesis (that is, pantetheinyl aminoacyl-thiol $1^a_{Aaa}$) and, from there, selective formation of aminoacyl nucleotides (**17**) is observed in water at neutral pH. The emergence of the biochemical intermediates of both ribosomal and non-ribosomal peptide synthesis from one chemical route, in water and without protecting groups, is distinct and unexpected. Coupling nitrile reactivity with the physical (eutectic) behaviour of phosphate solutions[41] makes aminoacyl-thiol synthesis prebiotically plausible and warrants further (geochemical) investigation of this process in model (arctic or polar) soda lakes that are known to accumulate high concentrations of phosphate[44].

We have discovered that aminoacyl-thiols **1** and biological thiol cofactors (for example, pantetheine **5a** and coenzyme M **5c**), make RNA aminoacylation a predisposed chemical process in water. The formation of aminoacyl-RNAs (**17**) is the first step of RPS but had proved to be extremely challenging under prebiotically realistic conditions[4,5]. Thioester-mediated RNA aminoacylation resolves this problem, occurring with unprecedented efficiency and side-chain tolerance in water at neutral pH to yield the substrates that are required for RPS in water. Our results indicate that thioesters (and thiol cofactors) may have played a key role during the early biochemical evolution of protein synthesis. Although the de novo development of a 'ribosome' and the iteration of the second step of protein synthesis are beyond the scope of the present work, we have demonstrated that a simple switch in acyl activation, from thioester (**1**) to thioacid (**11**) activation, switches on peptide synthesis and switches off diol aminoacylation. Together, these reactions demonstrate that the two steps of RPS can be chemically orchestrated without evolved catalysts. Future work will focus on developing peptidyl-RNA coupling to enable iteration and catalytic control over the second step of RPS.

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

# Methods

## General nucleoside aminoacylation protocol

The specified aminoacyl-thiol ($1_{Aaa}^{a-e}$) and pentaerythritol (internal standard) were dissolved in degassed $H_2O/D_2O$ (98:2) at the specified pH and added to the specified nucleoside or nucleotide (**16**, **20**–**32**). The pH of the resultant solution was measured and, if necessary, corrected to the stated pH with HCl/NaOH and then this solution was diluted with degassed $H_2O/D_2O$ (98:2) to give the specified reactant concentrations. The solution was then incubated at room temperature and nuclear magnetic resonance (NMR) spectra were periodically acquired.

## General nucleic acid aminoacylation protocol

The specified gel-purified FAM-oligonucleotide (**ON**#; 10 μM, 0.5 μl), +/−10-mer or 15-mer complement **ON** (10 μM, 0.6 μl) and +/−5-mer downstream **ON** (10 μM, 0.6 μl) were added to the reaction buffer (2 μl, 2.5 M KCl, 1–2 M MES, pH 6.5). The specified thioester ($1_{Aaa}^{a-e}$; 400 mM, 5 μl, pH 6.5) was then added and the solution diluted to 10 μl with $H_2O$. The resultant solution (which contained 0.5 μM RNA substrate, ±0.6 μM 10-mer or 15-mer-complement, ±0.6 μM 5-mer, 500 mM KCl, 200–400 mM MES pH 6.5 and 200 mM thioester) was vortexed, centrifuged and incubated at room temperature. At each required time point, an aliquot (1–2 μl) was removed. If the reaction contains ds-RNA, a competitor oligomer (30 equiv. with respect to FAM oligonucleotide) was then added to the aliquot and it was incubated for 5–10 min leading to the formation of a single-stranded FAM-tagged oligomer. Then a quenching buffer (16 μl; 10 mM EDTA, 150 mM NaOAc and 93% (vol/vol) formamide) was added to the aliquot and then it was analysed by polyacrylamide gel electrophoresis (PAGE; 36% wt/vol urea, 20% (19:1) acrylamide, 10% 10 × EA) in 1 × EA buffer (100 mM NaOAc, 2 mM EDTA, pH 5.2) pre-cooled to 1 °C. The gel was then shaken in 1 × TBE buffer (89 mM Tris, 89 mM borate, 2 mM EDTA, pH 8.3) for 5–10 min and the fluorescent (473 nm excitation) bands were quantified.

## General nucleoside peptide synthesis protocol

The specified aminoacyl-thiol ($1_{Aaa}^{a-e}$; 100 μmol) or **33** (40 μmol) was dissolved in degassed $D_2O$ (0.4 ml) or $H_2O/D_2O$ (98:2, 0.4 ml) +/− MES buffer (0.5–2.0 M) and the solution was carefully adjusted to pH 6.0–6.5 with NaOH or HCl as required. The specified nucleoside or nucleotide (**16** or **31**; 10 μmol) was then added. The solution was adjusted again to the specified pH value and diluted to 0.5 ml with degassed $D_2O$ or $H_2O/D_2O$ (98:2). The solution was then incubated at room temperature and NMR spectra were periodically acquired over 2–24 h. If MES buffer was not present in solution, MES buffer (1 M, 500 μmol, pH 6.5) was then added. $K_3Fe(CN)_6$ (150 or 450 μmol) and thioacid (**11**; 50 or 150 μmol) were then added, and after 10 min, the reaction mixture was centrifuged. The supernatant was incubated at room temperature and NMR spectra were acquired.

## Data availability

Data generated and analysed during this study, including experimental data, spectroscopic and PAGE images, are included in this article and its Supplementary Information.

**Acknowledgements** We thank the Engineering and Physical Sciences Research Council (EP/X011755/1 to M.W.P.), the Simons Foundation (1154101 to M.W.P.) and the Royal Society (URF\R1\231450 to D.W.) for financial support and A. E. Aliev, J. Attwater, K. Karu (UCL, Chemistry) and F. Werner (UCL, Structural and Molecular Biology) for technical support, access to equipment or facilities and helpful discussions.

**Author contributions** J.S., B.T., D.W. and M.W.P.: conceptualization, methodology and analysis. J.S., B.T., D.W., M.S.W., Y.Y. and M.W.P.: investigation. J.S., B.T. and D.W. contributed equally to the development of RNA aminoacylation. J.S. and M.S.W. developed peptidyl-RNA synthesis. D.W. and M.W.P.: acquisition of funding. M.W.P.: supervision and project management. J.S., B.T., D.W. and M.W.P. wrote the paper and all authors approved the final submission.

**Competing interests** The authors declare no competing interests.

**Additional information**
**Correspondence and requests for materials** should be addressed to Matthew W. Powner.

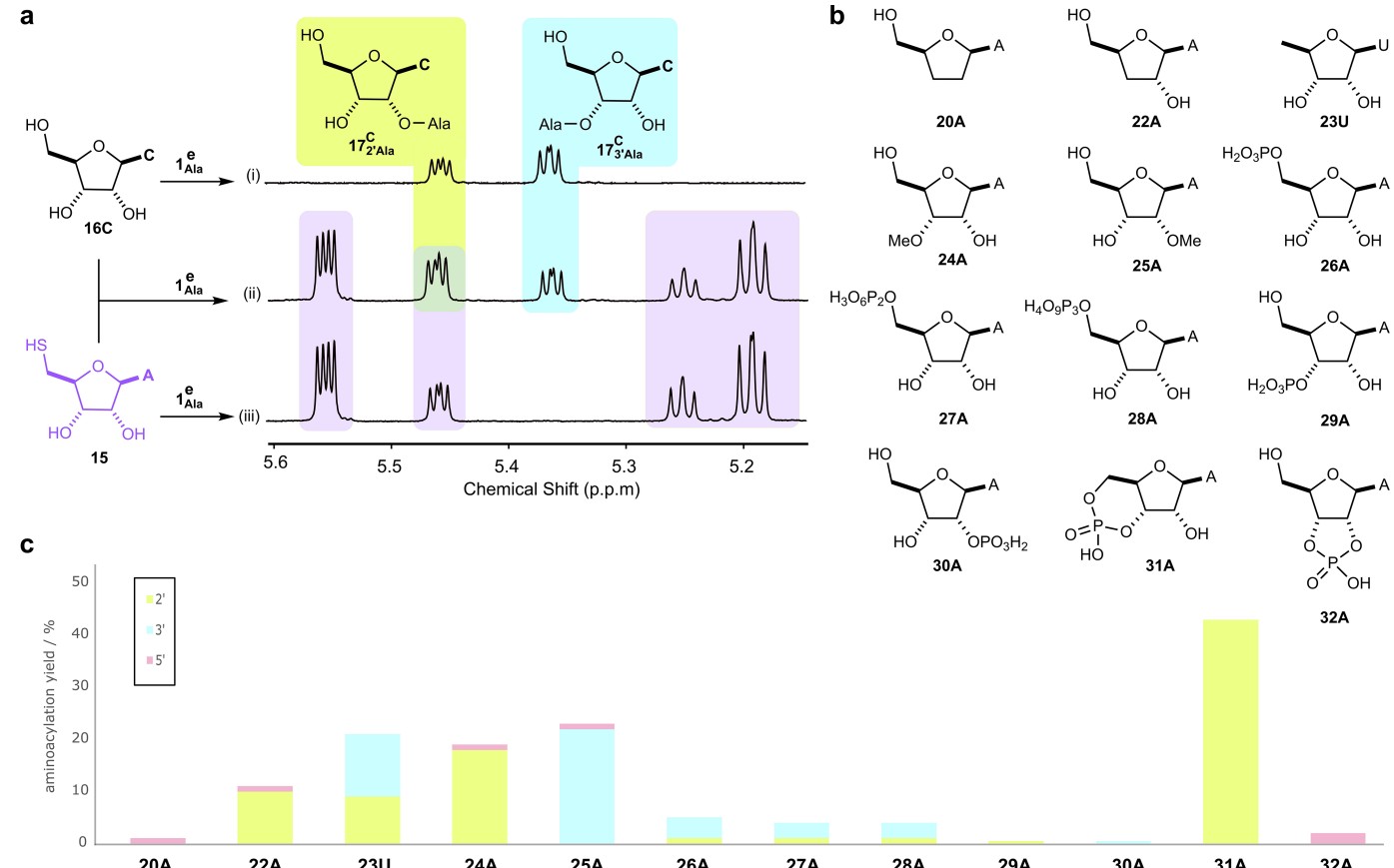

**Extended Data Fig. 1 | Reaction of alanyl-thiol with nucleosides. a**, ¹H NMR spectra show the reaction of: (i) L-1$^{e}_{Ala}$ (200 mM) and cytidine **16C** (20 mM) at pH 6.5, which yields aminoacyl-cytidine **17$^{C}_{2'Ala}$** (25%) after 24 h; (ii) L-1$^{e}_{Ala}$ (100 mM), **16C** (20 mM) and 5′-mercapto-adenosine **15** (15 mM) at pH 6.5 after 24 h; and (iii) L-1$^{e}_{Ala}$ (100 mM) and **15** (20 mM) at pH 6.5, in which 2′,3′-diol aminoacylation (54%) is observed after 24 h. **b**, Structure of non-canonical nucleosides (**20**, **22**–**25**) and ribonucleotides (**26**–**32**). **c**, Alcohol-aminoacylation yield for specified nucleosides (20 mM) with L-1$^{e}_{Ala}$ (200 mM) at room temperature and pH 6.5 after 24 h. See Fig. 2c for canonical RNA and DNA nucleosides. For more experimental details, see Supplementary Figs. 48–69 and Supplementary Table 6.

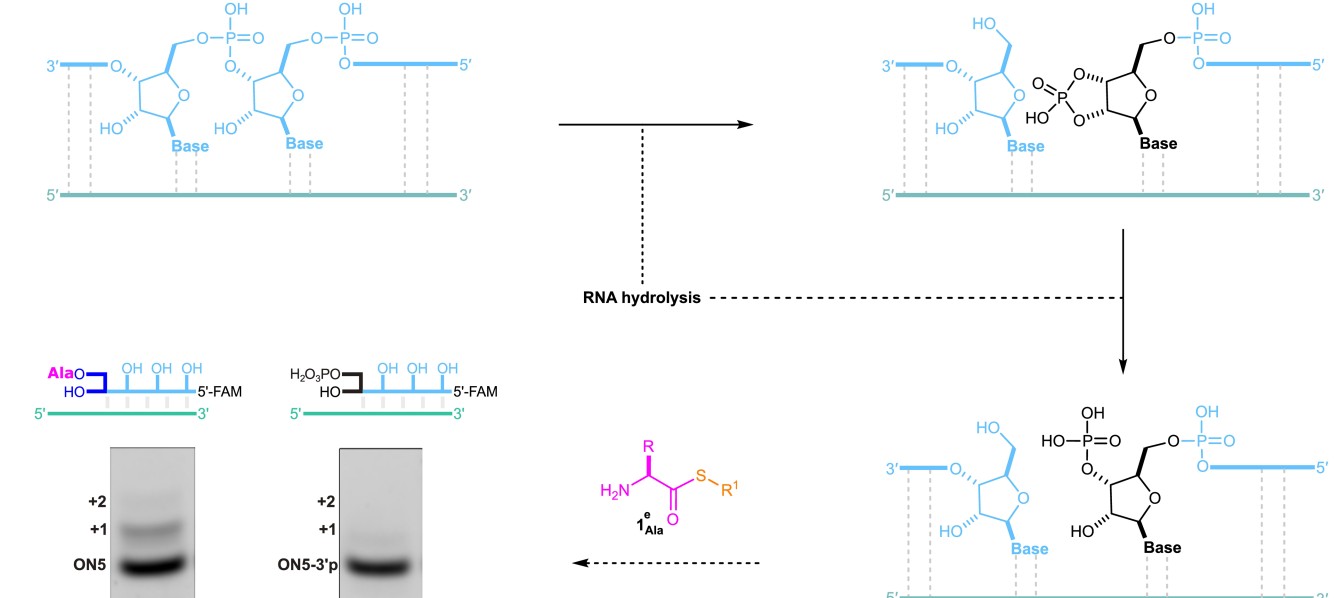

**Extended Data Fig. 2 | RNA hydrolysis suppresses adventitious aminoacylation.** RNA hydrolysis proceeds by means of fragmentation, which yields 2′-phosphate and 3′-phosphate through 2′,3′-phosphate intermediate. The high p$K_a$ of these alcohols inhibits aminoacylation and prevents the adventitious aminoacylation of RNA degradation products. PAGE images show the reaction of ds-oligomeric RNA (0.5 µM) with aminoacyl-thiol L-$1^e_{Ala}$ (200 mM, R = Me, $R^1$ = Et) in MES buffer (200 mM, pH 6.5) and KCl (500 mM) after 16 h at room temperature: left, 5′-FAM-UAGGAGACCA (**ON5**) + 5′-GCAGUUGGUCUCCUA (**ON7**); right, 5′-FAM-UAGGAGACCA-3′p (**ON5-3′p**) + **ON7**.

**a**

| Entry | 4 (mM) | 5c (mM) | pH | 1c (%) |
|---|---|---|---|---|
| 1 | L-Ala (20) | 80 | 3 | 29 |
| 2 | L-Ala (20) | 80 | 4 | 70 |
| 3 | L-Ala (20) | 80 | 5 | 81 |
| 4 | L-Ala (20) | 80 | 6 | 63 |
| 5 | L-Ala (600) | 1000 | 5 | 71 |
| 6 | L-Leu (1) | 2 | 6* | 73 |
| 7 | L-Leu (1) | 5 | 6* | 88 |
| 8 | L-Leu (1) | 10 | 6* | >95 |

| Entry | 4 (mM) | 5c (mM) | pH | 1c (%) |
|---|---|---|---|---|
| 9 | Gly (20) | 100 | 5 | >95 |
| 10 | L-Val (20) | 100 | 5 | >95 |
| 11 | L-Phe (20) | 100 | 5 | >95 |
| 12 | L-Leu (20) | 100 | 5 | 88 |
| 13 | L-Glx (20) | 100 | 5 | 84 |
| 14 | L-Glu (20) | 100 | 5 | 82 |
| 15 | L-Asp (20) | 100 | 5 | 49 |
| 16 | L-Ser (20) | 100 | 5 | 81 |

**b**

**Extended Data Fig. 3 | Synthesis of aminoacyl-thiols from activated amino acids. a**, Yield of aminoacyl-thiol $1^c_{Aaa}$ from NCA (**4**) and coenzyme M (**5c**) in MES buffer (200 mM) at room temperature after 10–30 min. *Phosphate buffer. Glx = glutamine-γ-nitrile. For further data and other thiols (**5**), see Supplementary Figs. 132–151. **b**, The reaction of aminoacyl-adenylate **2** (20 mM) with thiol **5b** (100 mM) in MES buffer (200 mM, pH 6.0) yields aminoacyl-thiol $1^b_{Ala}$ (42%). For yield of $1^b_{Ala}$ observed from reaction of aminoacyl-adenylate **2** and thiol **5b** at alternative pHs (pH 2.0–5.0), see Supplementary Table 39.

**a   aminoacyl-thiol formation**

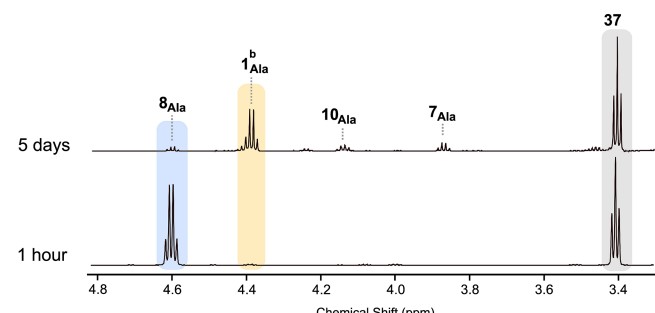

**b   α > β aminoacyl-thiol selective**

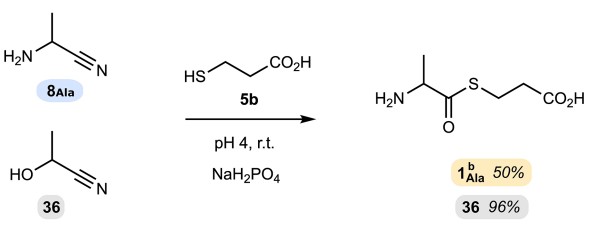
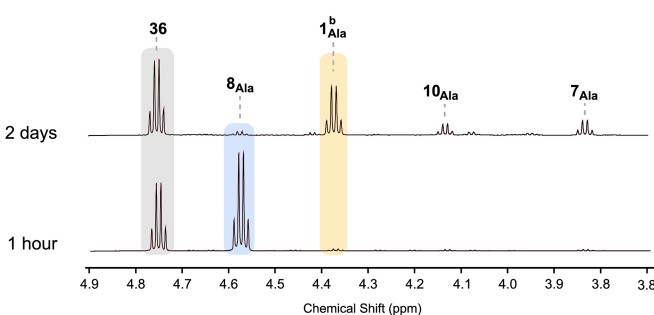

**c   amine > alcohol selective**

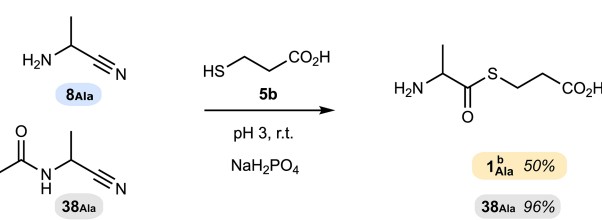

**d   amine > amide selective**

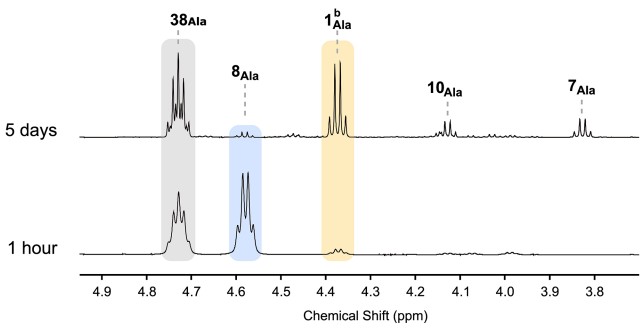

**Extended Data Fig. 4 | Synthesis of aminoacyl-thiols in water. a–d,** ¹H NMR spectra show the selective synthesis of alanine thioester **1ᵇ_Ala** (50–55%) from alanine nitrile **8_Ala** (200 mM) and thiol **5b** (9 equiv., 1.8 M) at pH 3.0 and 5.0 and room temperature, in the presence of: lactonitrile (**36**) (**b**); β-alanine nitrile (**37**) (**c**); and N-acetyl alanine nitrile (**38_Ala**) (**d**). See Extended Data Fig. 5 for low-concentration, neutral pH reactions.

**a**

| Entry | 8 | 5 ($R^1$) | buffer | pH | Time (days) | 1 (%) |
|---|---|---|---|---|---|---|
| 1 | Ala | **5b** ($CO_2H$) | - | 7 | 1 | n.d. |
| 2 | Ala | **5b** ($CO_2H$) | - | 5 | 1 | **20** |
| 3 | Ala | **5b** ($CO_2H$) | - | 4 | 2 | **42** |
| 4 | Ala | **5b** ($CO_2H$) | - | 3 | 5 | **48** |
| 5 | Ala | **5b** ($CO_2H$) | - | 2 | 5 | **28** |
| 6 | Ala | **5b** ($CO_2H$) | CB | 3 | 5 | **47** |
| 7 | Ala | **5b** ($CO_2H$) | PB | 3 | 5 | **56** |
| 8 | Ala | **5c** ($SO_3H$) | PB | 3 | 2 | **52** |
| 9 | Ala | **5a** pantetheine | PB | 3 | 5 | **39** |
| 10 | Ala | **5d** (NHAc) | PB | 3 | 2 | **41** |

**b**

| Entry | 8 | 5b (equiv.) | Time (days) | 1 (%) | Entry | 8 | 5b (equiv.) | Time (days) | 1 (%) |
|---|---|---|---|---|---|---|---|---|---|
| 1 | Ala | 1 | 7 | **12** | 11 | Gly | 9 | 5 | **46** |
| 2 | Ala | 2 | 7 | **28** | 12 | Leu | 9 | 18 | **53** |
| 3 | Ala | 5 | 1 | **9** | 13 | Pro | 9 | 3 | **64** |
| 4 | Ala | 5 | 2 | **17** | 14 | Phe | 9 | 10 | **21** |
| 5 | Ala | 5 | 3 | **24** | 15 | Val | 9 | 7 | **14** |
| 6 | Ala | 5 | 4 | **31** | 16 | Met | 9 | 7 | **38** |
| 7 | Ala | 5 | 7 | **43** | 17 | Arg | 9 | 3 | **41** |
| 8 | Ala | 7 | 7 | **52** | 18 | Ser | 9 | 3 | **40** |
| 9 | Ala | 9 | 2 | **45** | 19 | Lys | 9 | 7 | **44** |
| 10 | Ala | 9 | 7 | **54** | 20 | Pip | 9 | 5 | **58** |

**Extended Data Fig. 5 | Synthesis of aminoacyl-thiols from aminonitriles.**
**a**, The yield of alanyl-thiol $1_{Ala}$ from the reaction of alanine nitrile $8_{Ala}$ (0.2 M) and the specified thiol **5** (1.8 M, 9 equiv.) at specified pH and room temperature. PB, phosphate; CB, citrate; n.d., not detected. **b**, The yield of aminoacyl-thiol $1^b_{Aaa}$ from the reaction of specified α-aminonitrile **8** (0.2 M) and thiol **5b** (0.2–1.8 M, 1–9 equiv.) in PB at pH 3.0 and room temperature. Pip, pipecolic acid. An expanded dataset is provided in Supplementary Tables 41–46.

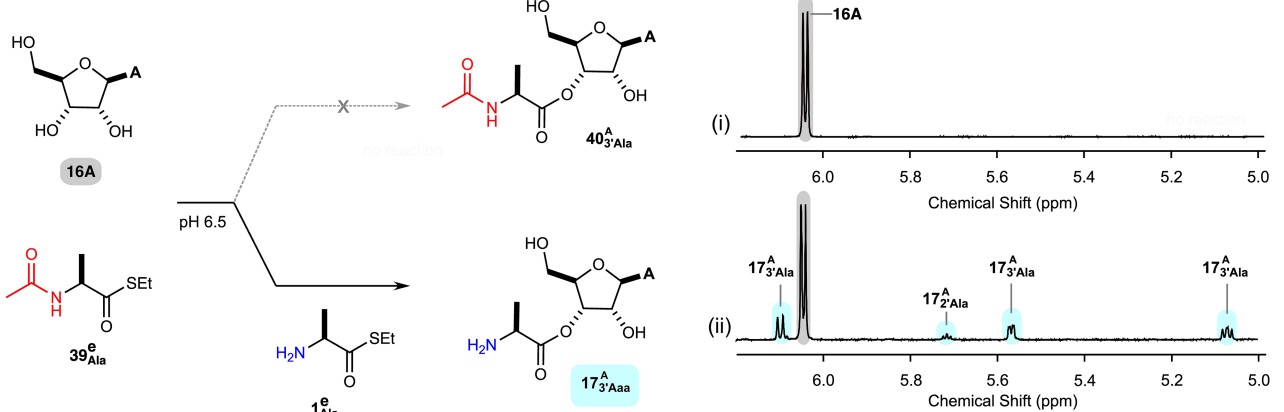

**Extended Data Fig. 6 | Diol acylation blocked by thioester-*N*-acylation.**
Stoichiometric competition between amidoacyl-thiol (**39**) and aminoacyl-thiol
(**1**) demonstrates the selective aminoacylation of RNA at neutral pH. Incubation
of adenosine **16A** (20 mM), *N*-acetyl-alanyl-thiol **39$^e_{Ala}$** (200 mM) and L-**1$^e_{Ala}$** (0 or
200 mM) in MES buffer (500 mM, pH 6.5) at room temperature yields: (i) with
no L-**1$^e_{Ala}$**, no nucleoside acylation and only unmodified **16A**; (ii) with 200 mM
L-**1$^e_{Ala}$**, aminoacyl-RNA **17$^A_{Ala}$** (20%).

**Extended Data Fig. 7 | Chemoselective two-step biomimetic peptide synthesis.** The reaction of adenosine **16A** (20 mM), *N*-acetyl-glycine thioacid **11**$_{Gly}$ (240 mM) and alanine thioester L-**1**$^e_{Ala}$ (240 mM) in MES buffer (500 mM, pH 6.5) at room temperature yields: (i) no amidoacyl-RNA, when $K_3Fe(CN)_6$ (600 mM) was added after 30 min; L-**1**$^e_{Ala}$ was the only amine after 30 min, so Ac-Gly-Ala-SEt (**39**$^e_{GlyAla}$) was observed as the only peptide product and **39**$^e_{GlyAla}$ does not react with **16A** (also see Extended Data Fig. 6); (ii) aminoacyl-RNA **17**$^A_{Ala}$ (21%) after 24 h; and (iii) **17**$^A_{Ala}$ formed in situ then furnishes peptidyl-RNA **40**$^A_{AlaGlyAc}$ (98%) in near-quantitative yield, when $K_3Fe(CN)_6$ (600 mM) was added after 24 h. For chemoselective one-pot peptidyl-RNA synthesis with Arg **33**, see Fig. 6c.

**Extended Data Fig. 8 | One-pot synthesis of peptidyl-RNA.** Yield of peptidyl-RNA **40** and **42** from the reaction of aminoacyl-RNA **17** or **41** with thioacid **11** (100–300 mM) and $K_3Fe(CN)_6$ (3 equiv.) in MES buffer (0.5–1.0 M, pD 6.0) at room temperature. All aminoacyl-RNAs were formed from their respective nucleoside (A, C, U = 20 mM, G = 2 mM) and thioester **1** (200 mM) or cyclic

arginine **33** (80 mM) and peptide synthesis implemented in situ, without isolation or any purification. *≈7:1 ε-$NH_2$/ε-NHGlyAc. ‡pD 6.5. For further experimental details, see Supplementary Tables 47–50 and Supplementary Figs. 229–263.

| Entry | aminoacyl-RNA | | peptide-thioacid 11 | | 40 |
|---|---|---|---|---|---|
| | **#B** | **Aaa** | **Aaa** | **(mM)** | **(%)** |
| 1 | **17A** | Arg | Gly | (105) | 93 |
| 2 | **17G** | Arg | Gly | (150) | 92 |
| 3 | **17U** | Arg | Gly | (200) | 97 |
| 4 | **17C** | Arg | Gly | (200) | 96 |
| 5 | **17A** | Ala | Gly | (300) | 90 |
| 6 | **17A** | Glu | Gly | (300) | 92 |
| 7 | **17A** | Leu | Gly | (350) | 95 |
| 8 | **17A** | Lys | Gly | (300) | 93* |
| 9 | **17A** | Ser | Gly | (300) | 91 |
| 10 | **17A** | Arg | Val | (300) | 88 |

| Entry | aminoacyl-RNA | | peptide-thioacid 11 | | 40 or 42 |
|---|---|---|---|---|---|
| | **#B** | **Aaa** | **Aaa** | **(mM)** | **(%)** |
| 11 | **17A** | Arg | Phe | (300) | 85 |
| 12 | **17A** | Arg | Met | (300) | 87 |
| 13 | **17A** | Arg | GlyGly | (110) | 92 |
| 14 | **17A** | Arg | AlaAla | (150) | 89 |
| 15 | **17A** | Arg | AlaPro | (200) | 82 |
| 16 | **17A** | Arg | MetGly | (120) | 77 |
| 17 | **17A** | Arg | GlyGlyGly | (200) | 96 |
| 18 | **41A** | Arg | Gly | (150) | 92‡ |
| 19 | **41A** | Arg | GlyGly | (110) | 93‡ |
| 20 | **41A** | Arg | GlyGlyGly | (100) | 95‡ |