## [Peer Review file · Nature]

Thioester-mediated RNA-aminoacylation and peptidyl-RNA synthesis in water.

Corresponding Author: Professor Matthew Powner

Version 0:

Reviewer comments:

Referee #1

(Remarks to the Author)

This paper describes a novel approach to RNA aminoacylation chemistry that operates through the attack of thiols on aminonitriles to generate aminoacyl thioesters, which in turn act to aminoacylate the cis-diol of nucleosides or RNAs. This approach addresses a key problem in early metabolism, and indeed the authors frame their work as part of the transition from prebiotic chemistry to metabolism and peptide synthesis. In order for aminoacylated RNAs to be useful for anything in a primitive cell, they would have to be made and used inside that cell, which means that the reaction conditions have to be mild and concentrations moderate. I cannot see how the reactions described in this paper, which are mostly done at a pH of 3, and extreme reactant concentrations (e.g. Fig. 1, aminonitriles at 0.2 M, and thiols at 1.8 M, with the thiol concentration not explicitly stated, and only referred to as 9 eq.), could possibly be relevant to cellular metabolic process. In the few experiments done at more moderate conditions, the yields are not any better than those achieved in the 1970s by Lohrmann, Orgel and Weber under similarly implausible reactant concentrations.

It is only at the very end of the paper that the authors suggest that a version of their chemistry might work with specific RNAs under more reasonable conditions if the RNA can provide a modest catalytic boost to the process. In my opinion the paper would be greatly improved if the authors framed their chemistry not as something that could be effective on its own, but as the potential basis for a series of RNA catalyzed metabolic transformations that use aminonitriles as the substrates for RNA aminoacylation. In its present state this paper would be more appropriate for JACS. I would recommend a transfer to Nature Chemistry if that journal wasn't so dysfunctional and unable to handle papers in a timely manner.

Referee #2

(Remarks to the Author)

Powner and co-worker report a chemical method, which allows them to activate carboxylic acids in water for the formation of esters. Importantly they can create esters in water with nucleosides. More importantly, the ester formation is chemoselective, as is occurs with high preferences only at the 2' and 3' OH groups. The ability to form selectively 2', 3'-ester of nucleosides is important in the context of the origin of life and here specifically for the question how RNA has once gained the ability to form peptide chains.

The chemistry behind the manuscript is straight forward. Nitriles are reacted with thiols, which generates under mildly acidic conditions (pH = 3-5) the corresponding thioesters. These react subsequently with other thiols to other thioesters or with alcohols to esters. Interesting is the fact that the reaction of amino-nitriles does not provide amides. Self-condensation does not occur and this allows the formation of activated amino-acid thioesters for subsequent ester formation with the 2', and 3'-OH groups of nucleosides or even RNA.

The authors describe a large number of interesting observations, but I did not find any explanation for the observed chemoselectivities. It seems that the amines just do not react because they are protonated at neutral pH, at which the authors perform the esterification reactions. Thiols and alcohols react because they are under pH neutral conditions good nucleophiles. It would be essential to learn at which pH value the amines start to react.

The stability of the formed thioesters is another important factor that is not discussed. Once the thioesters are formed in

water, the question is how long are they stable in a pH regime between 5 and 9, which was likely the prebiotic pH environment. This is a very important question because the formation of nucleoside esters in the presence of water is at first a very surprising result. Water and alcohols have quite similar reactivities, of course. The esterification reaction are performed under rather concentrated conditions and I believe this is done to obtain reaction rates that can compete with hydrolysis. Stability studies of the thioesters are consequently of utmost importance to judge how likely it is that the activated thioesters find the nucleosides before their hydrolysis.

The observation that the thioesters react preferentially with the 2',3'-OH groups and not with the primary 5'-alcohol at the nucleosides also requires further attention. No doubt, the primary 5'-OH groups is by far the kinetically more reactive alcohol. How do the authors explain the selective formation of the sek. 2', and 3' OH groups. Potential explanations are that the cis-diol structures offers a thermodynamic stabilization of the ester or (as briefly suggested by the authors) that the pKa values are lower for the sek. OH groups. In the first scenario the selectivity is thermodynamically driven, in the second case it is a kinetic phenomenon (which is hard to believe). Or, does in this case reaction occur first at the 5'OH groups followed by the transfer of the ester to the sek. OH positions?

In order to judge the prebiotic relevance, it would be essential to learn about the stability of the 2',3'-ester again under "normal" pH-conditions. Richert et al observed that these esters have only a very limited stability, which is generally insufficient for efficient peptide formation reactions.

This is maybe the most critical issue. The reported methods for the activation of amino acids and the formation of the nucleoside esters of amino acids is only valuable if the formed activated structures can be used to form peptides. While this was shown by Richert and others, the manuscript of Powner ends with the formation of just the activated esters. If peptides can not be formed, the whole activation story would become a bit meaningless. I therefore recommend to investigate first if the activation chemistry allows the formation of peptides with decent efficiency. The authors need to show that they can create peptides of a certain length with their chemistry before the importance of the thioester formation method and the observed chemoselectivities can be judged. In the current manuscript the activation chemistry is studied in great detail (without offering explanations for the observed chemoselectivities) but the value of the method has not been shown. Finally, I find the manuscript is overloaded with data to the different amino acids, which makes it hard to discover the key points. This over-complexity (particularly of the Figures) somehow blocks the ability to understand the key messages.

Version 1:

Reviewer comments:

Referee #1

(Remarks to the Author)

The authors have substantially revised and improved their paper. The selectivity for aminoacylation vs peptide formation is remarkable and is an important finding. I recommend publication of this paper. My one remaining reservation concerns the statements in the abstract about the selectivity of the aminoacylation reaction. While the aminoacylation chemistry is certainly selective for the cis-diol of RNA vs. over amines, it is not selective over internal 2'-hydroxyls in ss RNA (as shown in lines 216-219). The current phrasing on diol selectivity does not make this clear. It would therefore be better to shorten the abstract as follows:

From

...aminoacyl-thiols (1) react selectively to yield 2',3'-aminoacyl-RNA at neutral pH. 1 react selectively with RNA-diols over amine nucleophiles, promoting aminoacylation over adventitious (non-coded) peptide bond formation

To

...aminoacyl-thiols (1) react selectively with RNA-diols over amine nucleophiles, promoting aminoacylation over adventitious (non-coded) peptide bond formation.

The following minor comments (mostly typos) should also be addressed.

Line 112: typo, should be 'despite'

Line 298: what ribozymes?

Line 454: with peptide-thioester Ac-Ala-Set is not a peptide thioester, it is just an N-acetyl-aminoacyl ester

Line 469: underling should be underlying

Line 546: Should be orchestrated

Lines 573 and 575, should be adenylate

Referee #4

(Remarks to the Author)

This is a remarkable manuscript. Singh et al. demonstrate the selective aminoacylation of nucleoside and oligonucleotide in water with thioester, and the synthesis of elongated peptidyl-RNA from one-pot reactions between the resulting 2',3'-

aminoacyl-riboside with peptide-thioacid. The work strikes the right balance in selective activation chemistry to properly funnel reactants down paths that lead to the synthesis of peptides in a ribonucleoside dependent fashion, as found in extant biology. Along the way, the authors demonstrate how thioesters can be generated from N-carboxyanhydrides and thiols, how duplexes of RNA can guide aminoacylation to the terminus of RNA, and how a terminus similar to that of tRNA facilitates aminoacylation. The work is quite thorough and could be split into multiple papers, although I would not advise doing so. In the current manuscript, a logical, easy to follow, complete story is presented that will, in my opinion, make a strong, long-lasting impact on the prebiotic chemistry field.

I have gone through the comments from review 2 and find all of the responses thorough and convincing. The authors made appropriate changes to the manuscript and have now even demonstrated coupled peptide bond formation.

I have no interest in suggesting more experiments. There is already more than enough there to make a convincing story (and the manuscript already has 284 pages of supplementary information).

I do have a few comments that the authors may wish to consider, but again, I am NOT requesting more experiments.

The authors note that the thioester of Arg was particularly effective in aminoacylation and attribute this activity to the formation of a cyclic intermediate. That may very well be the case, but the known ability of Arg to interact quite effectively with RNA seems to be overlooked. Is it not reasonable to expect that a molecule that has greater intrinsic ability to interact with RNA would also aminoacylate RNA more effectively? RNA-binding proteins are typically enriched in Arg, and Lys does not interact as extensively with RNA as Arg.

I'd also like to note that the guanidine/guanidinium group of Arg is not shown protonated in figure 4, and I'd expect that the protonated form would dominate at the pH of the experiments. A justification for this in the text seems lacking.

Am I understanding correctly that the ability of Cys to react with amino acid thioester to form a dipeptide is not problematic in the scheme proposed by the authors? Wouldn't the resulting dipeptide be capable of further extension with thioacid, as the Cys would no longer reside at the amino-terminus? That is, extension could continue with the residue at the N-terminus and the now more internal Cys would not continue to appreciably react with thioesters? If that's the case, perhaps that's worth mentioning.

There are nucleotides, such as NAD(H), that are important for metabolism. I wonder if thioesters would efficiently aminoacylate such nucleotides, and what the effect of such activity would be. Again, I'm not requesting that this experiment be run or even that the authors comment on this in the manuscript. This is more a comment to consider (or not) for future studies.

The authors suggest "(arctic or polar) soda-lakes" as a potential geological setting. I know that it is difficult to say with much confidence where the described chemistry could have taken place, but I think it is useful for the field to actively think of plausible settings. As the proposed chemistry exploits many sulfur-containing molecules (and also exploited iron as $K_3Fe(CN)_6$), there are clearly more constraints on what would have been necessary. There are also many other papers from the Powner group and colleagues that are related and necessary for the described scenario. What do all of these data tell us regarding a plausible setting?

I have to say that I have grown tired of reading exciting titles that then let me down when I read the actual work described in the paper. I felt the opposite with this manuscript. The title was intriguing, of course, but my enjoyment only increased as I continued to read the paper. It's very nice work.

We would like to begin by thanking our reviewers for their positive appraisal of the novelty and importance of our aminoacylation, as well as for their time and constructive comments.

We have addressed all their comments in the point-by-point response below. We have also made several major changes and significant additions to our original manuscript.

The major changes are listed below:

List of major changes to manuscript:

1. We have changed the manuscript title to reflect the (major) new addition to our work, see addition #12 (peptide synthesis).
2. An additional author, Max Satterly Webley (MSW). Max joined the project to develop peptidyl-RNA synthesis, and the new section added to our manuscript.
3. The key result in our paper was that under mild (neutral pH) conditions we observed unprecedented selectivity for nucleoside-diol, over amine, aminoacylation by biological aminoacyl-thiols (**1**). We have rewritten and restructured the whole manuscript and all the figures so that this aminoacylation is presented as the primary result and at the outset of the paper.
4. Additionally, we have added an overview scheme, new Figure 1, to clarify that:
(i) the pH of aminoacylation reaction was optimal at neutral pH (pH 6–7), and the vast majority of reactions reported in our paper are at pH 6.5;
(ii) the substrates in aminoacylation reaction, (i.e., thiols (**5**), aminoacyl-thiols (**1**) and RNAs) are all biological and all can be found in modern cells today on Earth;
(iii) the key discovery is that aminoacyl-thiols (**1**) chemo-selectively achieve the first step of ribosomal peptide synthesis, that is the selective aminoacylation of RNA in water at neutral pH.

Our new figure aims to clearly show that aminoacylation of RNA is orthogonal to peptide synthesis due to the nature of (biological) thioester activation. Selectively achieving RNA-aminoacylation (without synthetase enzymes) provides a non-enzymatic mechanism to bypass the chicken-and-egg problem that is inherent to protein synthesis, i.e. proteinaceous synthetases are required for protein synthesis in biology. We demonstrate how the reactivity of aminoacyl-thiols (**1**) allows the two steps of biological peptide synthesis to be orchestrated and controlled, without evolved catalysts, solely through the natural reactivity of simple biological molecules.

Please see addition #12 below for newly added one-pot peptidyl-RNA synthesis in water at neutral pH and clear demonstration of unprecedented chemical control over both RNA-ester and peptide synthesis in water at the same pH.

5. We have added (to the new Figure 1) an additional graph to pictorially demonstrate the **selectivity of aminoacylation vs peptide** synthesis under optimal conditions (at pH 6.5). See Fig. 1c.

6. We have added a graph (to Figure 2) to show the **stability of aminoacyl-thiols (1)** at pH 5–8, the reactivity at pH 9 and pH 10 are also additionally shown in Supplementary Figure 45 and 74.
7. We have added two new graphs to show the difference in RNA and DNA aminoacylation (see Figure 2C(iii)), and the reactivity of nucleotides and non-canonical nucleosides (see Extended Data Figure 1). Importantly, these findings provide clear evidence of **direct kinetic selectivity for RNA diol aminoacylation**.
8. We have added further data on the **high yielding synthesis of aminoacyl-thiols (1) at low (1 mM) concentration** in water, demonstrating the high-yielding synthesis of aminoacyl-thiol (1) at lower concentration. For example, 1 mM Leu-NCA (**4_{Leu}**) and 2 mM thiol (**5**) furnish aminoacyl-thiol (1) in 73% yield. See Extended Data Figure 3.
9. We have added an analysis of peptide formation from aminoacyl-thiols (1) at pH 9 and pH 10, as requested by reviewer #2. These high pH reactions are less effective for peptide synthesis than the pH 8 reactions that we had already presented. This is in line with our expectation because the pK_{aH} of the aminoacyl-thiol (1) is ~ 7.5 (see Supplementary Figure 47), and so little further deprotonation is expected upon increasing the pH from 8 to 9 or 10. We have now specified this pK_a in the manuscript. Peptide synthesis with aminoacyl-thiols (1) was worse, not better at higher pH, and our interest remains neutral pH reactivity where aminoacyl-RNA (**17**) is formed extremely selectively. Therefore, these new (high pH) data are only presented in the supplementary information (Supplementary Figure 45 and 74).
10. We have added data to demonstrate that **3'-phosphorylation blocks oligonucleotide aminoacylation**, and therefore adventitious aminoacylation due to RNA hydrolysis, see Extended Data Figure 2.
11. We have added data to the paper (see Figure 3c(i)) to demonstrate **2',3'-dideoxy-RNA** completely blocks oligonucleotide aminoacylation.
12. **Peptide synthesis** was originally beyond the scope of this paper. Developing (de novo) the reactivity achieved by the (2.5 million Dalton) ribosome complex (in addition to the biomimetic synthetase activity we have developed) remains, rightly and justifiably, beyond the scope of this paper which is about the selective synthesis of aminoacyl-RNA in water. However, that said, at reviewer #2's request we have now additionally demonstrated protecting-group-free, prebiotically plausible, peptidyl-RNA synthesis in water at neutral pH (see new Figure 6, and new Extended Data Figures 6, 7 & 8).

This new addition is, to the best of our knowledge, the **highest yielding and most selective chemical synthesis of peptidyl-RNAs** reported in the literature.

This new addition was a considerable body of additional work, including a large body of novel peptide synthesis reactions and corresponding syntheses of both aminoacyl-RNA and peptide-thioacid substrates that were required to undertake each of the peptidyl-RNA synthesis reactions.

Our aminoacylation also separately provides a missing piece to the puzzle under investigation by Szostak and coworkers, who have just published the statement that: "*due to the poor nucleophilicity of the 2',3'-diol of RNA, activated amino acids tend to either hydrolyze or polymerize into random peptides before RNA aminoacylation can occur*" [Proc. Natl Acad. Sci. U.S.A., **121** e2410206121, (2024)].

But much more importantly than any of these model chemical studies, our work provides a missing piece to the (real world) puzzle of the origins of translation and the origins of biological peptide synthesis: that is how to selectively synthesise aminoacyl-RNA in water. It is important that the missing piece of the puzzle are aminoacyl-thiols (**1**), because aminoacyl-thiols (**1**) are biological. Aminoacyl-thiols (**1**) are the key intermediates of non-ribosomal peptide synthesis in modern cells, and coenzyme A is the universal acyl-transfer factor across all forms of extant life.

To extend our paper further and demonstrate the in-situ formation of peptides from the aminoacyl-RNAs synthesised by thioester mediated aminoacylation, we have added a whole new section to our paper: "**Chemical differentiation of aminoacylation and peptide synthesis**". This section now demonstrates that the difference in reactivity between aminoacyl-thiol (**1**) and peptidyl-thioacids (**11**) provides the next piece of this puzzle: that is how to chemically control biomimetic peptide synthesis (on RNA) in water without evolved enzymes.

We demonstrate that peptide-thioacids (**11**) react selectively in water with aminoacyl-RNAs (**17**) to **yield peptidyl-RNAs (**40**) in near-quantitative yield and with remarkable selectivity and retention of chirality**. The switch in chemistry we demonstrate is remarkable. Thioacids do not react with nucleoside alcohols in this reaction, only with the free amine of the aminoacyl-RNA. This now achieves the second step of ribosomal peptide synthesis in an enzyme-free reaction in water at neutral pH.

This new study demonstrates that two different, but extremely closely related, modes of activation (i.e., thioester and thioacid activation) provide the reactivity required to orthogonally achieve the two steps of ribosomal peptide synthesis under the same reaction conditions (i.e., in water at pH 6.5).

We have demonstrated no product/intermediate purification is required during peptidyl-RNA synthesis; the high-yielding, one-pot (pH 6.5) conversion of nucleosides and nucleotides to peptidyl-nucleosides and peptidyl-nucleotides is demonstrated. The new peptide formation reaction added to the manuscript is observed to furnish excellent yields of peptidyl-RNA (up to 97% yield) and with complete selectivity (see Figure 6 and Extended Data Figure 6–8). Remarkably, due to the mild aqueous conditions, the stereochemistry of the aminoacyl-thiols (**1**), aminoacyl-nucleosides (**17**), and peptide-thioacids (**11**) was preserved over both steps. This is a striking result, starting from homochiral substrates and ending with a single isomer of peptidyl RNAs (**40**).

The reactions we report are uniquely controlled by the mode of acyl-activation; there is no need for evolved catalysts or intramolecular transfer or Watson-Crick templated reactivity. Thioesters are reactive with 2',3'-diols not amines, whereas activated thioacids react with amines (such as aminoacyl-RNA) and not 2',3'-diols. This provides a chemical

switch between aminoacylation and peptide synthesis under the same (pH and concentration) conditions.

This switch in reactivity is truly remarkable and will be of broad interest beyond the origins of life community. We know of no other chemical process or mechanism that can make this type of switch between the reaction of two different nucleophilic heteroatoms under the same reaction conditions.

Point-by-Point response to reviewer comments

Referee #1 (Remarks to the Author):

Comment 1A: *This paper describes a novel approach to RNA aminoacylation chemistry that operates through the attack of thiols on aminonitriles to generate aminoacyl thioesters, which in turn act to aminoacylate the cis-diol of nucleosides or RNAs.*

Response 1A: We thank reviewer #1 for highlighting the novelty of our approach to aminoacylation of RNAs with aminoacyl-thiols in water. But more importantly we thank reviewer #1 for demonstrating to us that we have poorly communicated the core message and key result in our manuscript.

The key result is the mild (neutral pH) conditions and unprecedented selectivity for alcohol over amine reactivity seen in the aminoacylation of nucleosides by biological aminoacyl-thiols.

This core message was reflected in the title of our paper, but we accept that we did not communicate this as well as we should have throughout the paper.

Thanks to the reviewers comment we now see that we had **incorrectly focussed on nitrile chemistry** at the start of the paper because of our own prior interest in nitrile reactivity, rather than focusing on the most important outcomes from a biological perspective.

We have generated aminoacyl-thiols from prebiotically plausible NCAs, biological aminoacyl adenylates and amino acid anhydrides, as well as from aminonitriles. However, the key result is the **selective reactivity of aminoacyl-thiols** rather than their synthesis.

We have now re-structured our manuscript so that aminoacylation is presented with the primacy it deserves and thank reviewer #1 for highlighting the flaw in our previous presentation of our results and the remarkable selectivity of aminoacyl-thiols.

Comment 1B: *This approach addresses a key problem in early metabolism, and indeed the authors frame their work as part of the transition from prebiotic chemistry to metabolism and peptide synthesis.*

Response 1B: We are delighted by the referee's characterisation of our work as addressing a key problem in early metabolism, prebiotic chemistry and peptide synthesis.

Comment 1C: *In order for aminoacylated RNAs to be useful for anything in a primitive cell, they would have to be made and used inside that cell, which means that the reaction conditions have to be mild and concentrations moderate.*

Response 1C: We thank reviewer #1 for this provocative thought.

Ideally, aminoacylation would occur under mild conditions (e.g. near neutral pH) and with high selectivity for the alcohols of nucleic acids over competing nucleophiles likely to be present in a cell (e.g. amines, phosphates, amino acid side chains). In fact, these are all features which are characteristic of the aminoacylation of nucleic acids by aminoacyl-thiols which we have discovered!

But it is incorrect to assume that all valuable compounds ‘must’ be made within the cell. This is clearly not true, even in a modern cell. Many compounds of biochemical value cross cell membranes; this would be especially likely with primitive membranes and protocells.

Whilst studying aminoacylation within a membrane (or primitive cell) is certainly extremely interesting, it is beyond the scope of our manuscript. However, **aminoacyl-thiols (1) are neutral species at neutral pH**, because the amino thioester $pK_{aH} = 7.5$ (see Supplementary Figure 47). In their neutral form these thioesters would be lipid soluble, as well as water soluble, and so would easily cross simple (prebiotic) membranes. Furthermore, due to the rapid and highly effective thiol exchange (that we have demonstrated – see Figure 2) lipophilic thiols or amphiphilic thiols, for example one related to the enzyme cofactor lipoic acid, could also promote aminoacyl-thiol transfer across membranes.

These are all extremely interesting concepts, but they are beyond the scope of this paper that is focused on the selective aminoacylation of RNA in water.

Comment 1D: *I cannot see how the reactions described in this paper, which are mostly done at a pH of 3, and extreme reactant concentrations (e.g. Fig. 1, aminonitriles at 0.2 M, and thiols at 1.8 M, with the thiol concentration not explicitly stated, and only referred to as 9 eq.), could possibly be relevant to cellular metabolic process.*

Response 1D: We thank reviewer #1 for highlighting that we had poorly communicated the mild conditions of our aminoacylation of RNAs, and we believe that our new manuscript clarifies the mild, neutral pH conditions of our aminoacylation.

(i) Reaction pH: Most reactions in our paper were carried out at pH 6–7 (see Figure 1, 2, 3, 4, 5 and 6) including, most importantly, the aminoacylation of nucleic acids and the (new) one-pot, protecting-group-free synthesis of peptidyl-RNA (see Figure 6).

We believe that this misconception, that most of our reaction were done at pH 3, was due to the poor way in which we had structured the manuscript. Our revised manuscript should make it much easier to see the mild reaction conditions which characterize our work. The new manuscript presentation should make it clear that the vast majority of the reactions were carried out at pH 6.5 throughout our manuscript.

We report only one reaction that was required to be at a pH between pH 3 and pH 5. This one reaction (aminoacyl-thiol formation from aminonitriles and thiols) was optimal at pH 3. We reported substrate scope for this reaction at its optimal pH; this is how chemical investigations

should be carried out. However, to defend our study of aminoacyl-thiol formation from aminonitriles some comment is needed.

It is not fair to judge even this one reaction as though we have not already recognised and resolved the challenge imposed by pH 3. We had stated in our paper that this low pH must be reconciled with prebiotic plausibility, and we have provided a prebiotically plausible mechanism and practical method to achieve this reactivity. We have demonstrated how simply **freezing a neutral pH phosphate solution can easily and plausibly overcome this perceived problem**.

Both neutral pH phosphate solutions and freezing conditions are widely accepted to be prebiotically plausible, and together they make aminoacyl-thiol synthesis highly effective, and render low pH (pH 3–5) highly plausible.

We demonstrate that freezing a very dilute, **pH 7 solution** of aminonitrile (2 mM) and thiol (10 mM) yields aminoacyl thiol. Upon melting this yields a pH 7 solution of aminoacyl-thiol in excellent yield, which could then enter (bio)synthetic pathways. This was discussed in detail in the paper (also see Figure 5d).

We have also demonstrated the aminoacylation of RNA in frozen conditions (see Figure 2d). Investigating further the transition from ice-synthesis to aminoacylation is beyond the scope of this paper but is of note that RNA chemistry can also be facilitated in ice eutectics. For example, see the extensive body of work in this area by the Holliger group (e.g. Attwater et al, *Ice as a protocellular medium for RNA replication Nat. Commun.* **2010**, 1, 76).

(ii) Use of ‘equivalence’: Regarding our choice of equivalents rather than concentration for the thiol – we chose this nomenclature (i.e., molarity of limiting reagent and equivalents of other reagents) based on all precedent from organic chemistry literature. We believe this was most appropriate and we think that it was clear. We have, though, now added the thiol concentration to Extended Data Figures 4–5 (formerly Figure 1).

The thiol molarity (concentration) was (and still is) recorded in the Supplementary Information, where the concentration had already been explicitly stated in each reaction scheme.

(ii) Thiol concentration: We have further investigated reactions using lower concentrations of thiols with aminonitriles and explored higher pH conditions. At pH 5, the reaction resulted in aminoacyl thiol formation at 21% yield within one day (see Extended Data Figure 5 and new Supplementary Tables 41–45).

In this context we found NCA (**4**) are superior to aminonitrile (**8**) in generating aminoacyl thiol at lower concentration. High thiol concentrations are not essential for aminoacyl-thiol formation, see Extended Data Figure 3a in which 80 mM thiol is sufficient to generate an aminoacyl-thiol in 81% yield from NCA in water.

We have now added further data to demonstrate the high-yielding synthesis of aminoacyl-thiol at even lower (1 mM) concentration; **1 mM Leu-NCA and 2 mM thiol furnish aminoacyl-thiol in 73% yield**. See Extended Data Figure 3a and new Supplementary table 32.

Comment 1E: *In the few experiments done at more moderate conditions, the yields are not any better than those achieved in the 1970s by Lohmann, Orgel and Weber under similarly implausible reactant concentrations.*

Response 1E: Our aminoacylation yields and **crucially selectivity** is a major improvement over anything Orgel and Weber have reported.

Orgel's reported results are shown below in "Fig. 4" which has been taken from reference 11 in our manuscript (Weber, A. L., & Orgel, L. E. Amino acid activation with adenosine 5'-phosphorimidazolide. *J. Mol. Evol.* **11**, 9-16 (1978)). We have provided alongside this "Fig. 4" data from our reactions with aminoacyl thiols at pH 6.5.

Fig. 4. A - C. (A) Formation of MepA esters of the amino acids: gln, pro, gly, phe, arg, ser, thr, and ala at ambient temperature. **(B)** Formation of MepA-ser and seryls erine. **(C)** Formation of MepA-gly and glycyglycine. Reaction solutions contained 0.1 M ImpA, 0.5 M MepA, 3 M imidazole (pH 5.8) and 0.5 M amino acid (gln and phe concentrations were 0.25 M because of the limited solubility of these amino acids)

The **differences in reactivity are striking** – even comparing alanine, Orgel's highest yielding amino acid, we observe 4 times the amount of aminoacylation with alanine-thioester. If, instead, we compare the highest yielding aminoacyl-thiol in our results (Arg-SEt; see graph below or Figure 1 in our manuscript), we see a **remarkable 15-fold increase in yield** compared with Orgel's results (76% vs 5%).

Much more importantly though, our results show that aminoacylation of nucleosides is **extremely selective** over formation of peptide – aminoacyl-thiols react selectively with alcohols rather than with amines (graph above of DKP vs aminoacylation). In sharp contrast to our results, Orgel’s data show dipeptide formation to occur about twice as efficiently as aminoacylation of his nucleotide mimics (See Orgel’s Fig. 4B and C, above). **Orgel did not demonstrate how RNA-aminoacylation can be achieved through a reaction that is orthogonal to peptide synthesis.**

It is fascinating and unexpected that the **aminoacyl-thiol reacts with the diol >> amine**, and therefore our results will be of broad interest (not just within the context of the origins of life).

Furthermore, we have also demonstrated **diol selectivity in RNA-duplexes**. RNA-duplex-induced selectivity is an extremely important aspect of our work, but this selectivity was **not demonstrated by Orgel and co-workers** (their ‘model’ substrate has no inter-nucleotide hydroxyl-moieties). We provide the first chemical mechanism to selectively access 3'-aminoacylation of polymeric nucleic acid — i.e., the substrates required for ribosomal peptide synthesis.

There is also a **striking difference in longevity** for aminoacyl-RNA products under our reaction conditions compare to those reported by Orgel. Orgel’s yields all fall after 6 hours, whereas under our conditions the concentration of aminoacyl-RNA remains stable for days (see above for 40 hours at pH 6.0 with the alanine aminoacyl-thiol and 20 hours at pH 6.5 with arginine aminoacyl-thiol).

To quote Jack Szostak, writing last year about Orgel’s work and the method that Orgel employed [see *Proc. Natl Acad. Sci. U.S.A.*, **121** e2410206121, (2024)] “*activated amino acids tend to either hydrolyse or polymerise into random peptides before RNA aminoacylation can occur*”. This neatly summarises the current opinion in the field today, and why our results showing the opposite to be true are so unexpected and important.

Comment 1F: *It is only at the very end of the paper that the authors suggest that a version of their chemistry might work with specific RNAs under more reasonable conditions if the RNA can provide a modest catalytic boost to the process. In my opinion the paper would be greatly improved if the authors framed their chemistry not as something that could be effective on its own, but as the potential basis for a series of RNA catalyzed metabolic transformations that use aminonitriles as the substrates for RNA aminoacylation.*

Response 1F: We agree with referee #1 that the observation of RNA-catalysed aminoacylation (catalysed by the simple RNA-overhang (**ON10** + **ON13**) shown in Figure 4c), is an interesting element of our work. This observation opens the door to further study of how elements of RNA structures can control aminoacylation, perhaps ultimately in a coded manner. We have already explicitly laid out the case for this possibility and acknowledged the potential value in our manuscript (see pages 10–11).

Comment 1G: *In its present state this paper would be more appropriate for JACS. I would recommend a transfer to Nature Chemistry if that journal wasn't so dysfunctional and unable to handle papers in a timely manner.*

Response 1G: We hope that the new manuscript has put the correct emphasis on our key discoveries.

To further augment our results, and to further demonstrate the remarkable selectivity of aminoacyl-thiols, we have now additionally demonstrated the one-pot synthesis of peptidyl-RNAs in water at neutral pH.

Our new peptidyl-RNA synthesis is controlled by a uniquely simple switch in carbonyl-activation and builds directly on our thiol mediated RNA-aminoacylation.

To our knowledge the mode of selectivity and switch in reactivity is totally unprecedented. It provides the chemical selectivity required to render the two steps of ribosomal peptide synthesis orthogonal under the same (pH, concentration) conditions. This is a significant addition to our manuscript.

We thank the referee for highlighting the previous weaknesses in our presentation of our results, demonstrating the value of good peer review.

Referee #2 (Remarks to the Author):

Comment 2A: *Powner and co-worker report a chemical method, which allows them to activate carboxylic acids in water for the formation of esters. Importantly they can create esters in water with nucleosides. More importantly, the ester formation is chemoselective, as it occurs with high preferences only at the 2' and 3' OH groups. The ability to form selectively 2'-, 3'-ester of nucleosides is important in the context of the origin of life and here specifically for the question how RNA has once gained the ability to form peptide chains.*

Response 2A: We thank referee #2 for their complementary characterisation of our work.

We agree that the formation of esters in water is important, and we also agree that the selectivity that we have demonstrated for the 2',3'-diol of RNA is even more important.

We are grateful for reviewer #2's confirmation that the selective formation of aminoacyl RNA is important in the context of the origins of life.

Comment 2B: *The chemistry behind the manuscript is straight forward. Nitriles are reacted with thiols, which generates under mildly acidic conditions (pH = 3-5) the corresponding thioesters.*

We appreciate that referee #2 has characterised our conditions for thioester formation from nitriles as “*mildly acidic*”.

Comment 2C: *These react subsequently with other thiols to other thioesters or with alcohols to esters. Interesting is the fact that the reaction of amino nitriles does not provide amides. Self-condensation does not occur and this allows the formation of activated amino-acid thioesters for subsequent ester formation with the 2'-, and 3'-OH groups of nucleosides or even RNA.*

Response 2C: We agree wholeheartedly with referee #2 that the poor reaction of amino thioesters with amines is interesting.

Amino thioesters are remarkably unreactive to all amine nucleophiles tested (Figure 2a).

As noted by reviewer #2, this chemoselectivity for alcohols over amines goes in-tandem with the 2'/3'-diol over 5'-alcohol selectivity. Together these allowed us to demonstrate selective aminoacylation of RNA-diols in water, and in the presence of (a large excess of) amines.

Comment 2D: *The authors describe a large number of interesting observations*

Response 2D: We thank reviewer #2 for this complementary comment.

At the reviewers request we have now additionally provided even more observations in our revised manuscript to demonstrate the chemo-selective and high-yielding synthesis of peptidyl-RNA in water, under the same condition that RNA was aminoacylated. We demonstrate this can be chemically controlled by the subtle difference between thioester and thioacid activation in water at pH 6.5.

Comment 2E: *I did not find any explanation for the observed chemoselectivities. It seems that the amines just do not react because they are protonated at neutral pH, at which the authors perform the esterification reactions. Thiols and alcohols react because they are under pH neutral conditions good nucleophiles.*

Response 2E: We are grateful that referee #2 has highlighted one of the most interesting aspects of our work – the selectivity for aminoacylation of RNA over reaction with amines.

However, protonation at neutral pH is not as important as the reviewer suspects, firstly the pK_{aH} of aminothioesters is ~ 7.5 , and so they are mostly deprotonated at pH 7–8, but our reactions remain diol selective. Even aminonitriles ($pK_{aH} \sim 5$) are unreactive with thioesters at neutral pH.

As can be seen below in “*Fig. 4*” from Orgel’s work (Weber, A. L., & Orgel, L. E. *Amino acid activation with adenosine 5'-phosphorimidazolide. J. Mol. Evol.* 11, 9–16 (1978)) the selectivity that we observed is not at all inevitable. In **Orgel’s reaction peptide formation was observed to be the major product**, even though the pH was slightly more acidic than our conditions. As we have stated in the paper, we believe the selectivity is **due to the judicious balance between weakly activated electrophile and low pK_a alcohol nucleophilicity**.

Fig. 4. A - C. (A) Formation of MepA esters of the amino acids: gln, pro, gly, phe, arg, ser, thr, and ala at ambient temperature. **(B)** Formations of MepA-ser and serylserine. **(C)** Formation of MepA-gly and glycyglycine. Reaction solutions contained 0.1 M ImpA, 0.5 M MepA, 3 M imidazole (pH 5.8) and 0.5 M amino acid (gln and phe concentrations were 0.25 M because of the limited solubility of these amino acids)

Furthermore, in our newly added section on peptidyl-RNA formation we show that the amine moiety of aminoacyl-RNA is competent to react as a nucleophile at pH 6.5 – see Figure 6 in our new manuscript, which is also shown below – these are the same conditions under which aminoacylation with thioesters is diol-selective. So, the selectivity is not due to the protonation state of the amine but to the specific nucleophile-electrophile pairing.

Comment 2F: *It would be essential to learn at which pH value the amines start to react.*

Response 2F: We agree that the effect of pH on the self-condensation is an important consideration, this is something that we had already highlighted in our manuscript. We noted that:

“Peptide formation from **1** was ineffective at all pHs, and the major pathway was the slow hydrolysis of **1**. For example, alanine thioester L-**1**^e_{Ala} reacts sluggishly at pH 7 to afford a very low yield of diketopiperazine (**6**_{AlaAla}, <3%), alongside alanine (**7**_{Ala}, 48%) after 5 days (Fig. 2a). More DKP **6** was observed at higher pH, but the maximum DPK yield (only 21%) was observed at pH 8 (Supplementary Figure 45) and hydrolysis dominated at pH 9–10 (Supplementary Figure 74). Importantly, peptide formation was further suppressed at lower pHs (Fig. 1bi & Supplementary Figure 45).”

We had reported pH 8, and we continue to highlight pH 8 in the revised manuscript, because pH 8 led to the most peptide synthesis observed (albeit still a poor reaction).

We have now repeated our analysis at pH 9 and pH 10 these remain in the supplementary information because they are less effective for peptide synthesis than pH 8. Whilst the stability of aminoacylated RNA and aminoacyl-thiol is decreased further at pH 9–10, the self-reaction (peptide) yield does not increase, rather, more hydrolysis is observed at higher pH. This is in line with our expectation because the pK_{aH} of the aminoacyl-thiol is 7.5, and so little further

deprotonation is expected upon increasing the pH from 8 to 9 or 10. This additional data is presented here below (and has been added to our Supplementary Figure 74).

Supplementary Figure 74: with pH 9 and 10 data: Alanyl diketopiperazine 6_{AlaAla} formation (%) as determined by 1H NMR spectroscopy plotted against time in the reaction of 17U (20 mM) with L-1^eAla (200 mM) in 1 M buffer with PET (20 mM) as an internal standard. pH 5, 6, 6.5: 1 M MES buffer. pH 7, 8: 1 M MOPS buffer. pH 9, 10: 1 M pyrophosphate buffer; L-1^eAla (190 mM).

For reference for how poor this peptide synthesis is, see the near-quantitative yield observed at pH 6.5 in Figure 6 and Extended Data Figure 8 in our new manuscript, via thioacid mediated peptidyl-RNA synthesis.

Together these two reactions (shown together in Figure 6 in our new manuscript) demonstrate that it is not amine protonation that curtails peptide formation, but the nature of acyl-activation. This is **remarkable, unprecedented, and unexpected, but extremely valuable** for the controlled synthesis of peptidyl-RNA.

Comment 2G: The stability of the formed thioesters is another important factor that is not discussed. Once the thioesters are formed in water, the question is how long are they stable in a pH regime between 5 and 9, which was likely the prebiotic pH environment.

Response 2G: We apologise to reviewer #2 for not making these results clearer. However, we have already included the stability of alanine thioester in Supplementary Table 1 at pH 6.5 and pH 7 over 120 hours.

At these pHs, small amounts of hydrolysis (11% at pH 6.5) and minimal self-reaction to form diketopiperazines (1% at pH 6.5) are observed after 120 hours.

At alkaline pHs, the amino thioesters hydrolyse more rapidly, but at more acidic pHs they are more stable. However, operating in a regime near neutral pH, amino thioesters are stable over multiple days.

Importantly, the **aminoacyl thiols are clearly sufficiently stable to aminoacylate RNA** as has been demonstrated in our paper.

We have now added a graph to Figure 2b (see below) to demonstrate the stability of aminoacyl-thiols at pH 5, pH 6.5 and pH 8. Data at other pHs is provided in the supplementary information.

Comment 2H: *This is a very important question because the formation of nucleoside esters in the presence of water is at first a very surprising result. Water and alcohols have quite similar reactivities, of course. The esterification reactions are performed under rather concentrated conditions and I believe this is done to obtain reaction rates that can compete with hydrolysis. Stability studies of the thioesters are consequently of utmost importance to judge how likely it is that the activated thioesters find the nucleosides before their hydrolysis.*

Response 2H: We thank reviewer #2 for highlighting the importance of nucleoside ester formation in water.

We agree that this is indeed a very surprising result.

We also agree that understanding the rates of hydrolysis of both amino thioesters and nucleoside esters are important. This is why we have already plotted the rate of formation and subsequent hydrolysis of nucleoside esters in Figure 2c.

The plot in Figure 2 (below) shows that under alkaline conditions, hydrolysis of the nucleoside ester is rapid (Figure 2c(ii), blue circles). But they are stable for days (at pH 6.5) under the optimal conditions of our reaction (Figure 2c(ii), green diamonds).

Aminoacyl RNAs and aminoacyl thiols are both observed in water at pH 5–7 over the course of multiple days. We believe that the instabilities of these species have come to be (over) exaggerated in the literature through an (unjustified) focus on higher pH reactions. It is therefore **important that we demonstrate effective aminoacylation at pH 6–7, where these compounds are long lived in water.**

Comment 2I: *The observation that the thioesters react preferentially with the 2',3'-OH groups and not with the primary 5'-alcohol at the nucleosides also requires further attention. No doubt, the primary 5'-OH groups is by far the kinetically more reactive alcohol. How do the authors explain the selective formation of the sek. 2', and 3' OH groups. Potential explanations are that the cis-diol structures offers a thermodynamic stabilization of the ester or (as briefly suggested by the authors) that the pKa values are lower for the sek. OH groups. In the first scenario the selectivity is thermodynamically driven, in the second case it is a kinetic phenomenon (which is hard to believe). Or, does in this case reaction occur first at the 5'OH groups followed by the transfer of the ester to the sek. OH positions?*

Response 2I: We thank the referee for summarising why the selectivity we have observed is so interesting, and counter intuitive based upon 'first principles', and therefore why our results are so interesting and valuable.

The selectivity is clearly kinetic, contrary to reviewer #2's (reasonable) expectations. It is important of course that reviewer #2 notes (strongly and with an extended discussion of 'potential' alternatives) that this kinetic selectivity contradicts their expectations. We are most grateful to reviewer #2 for so lucidly outlining the widely accepted wisdom held to be true within organic chemistry that primary alcohols are more nucleophilic than secondary alcohols. And so, reviewer #2 states: "*No doubt, the primary 5'-OH groups is by far the kinetically more reactive alcohol*"; but our results demonstrate that there is more than doubt, our results demonstrate this statement is false.

Nucleotide secondary alcohols (including compounds **20A**, **23A** and **24A** which are secondary alcohols but not diols) are importantly **a special case**, due to the structure of the furanosyl ring to which they are attached. This is a very important point, that has been beautifully highlighted by

reviewer #2. It is likely an extremely important factor in the selection of RNAs for the role that they play in translation (as tRNA), and further emphasise the general importance of our finding.

The 2',3'-diol is kinetically more reactive toward the aminoacyl-thiol than either the primary 5'-alcohol or water: this can be clearly seen in our data. We believe this observation, and the fact that reviewer #2 find this “*hard to believe*”, both strongly underscores the fundamental importance of our study.

We have clearly demonstrated this in Figure 2c and discussed this selectivity in the main text. We have replaced the nucleoside scope table with two graphs that more clearly highlight the remarkable difference between:

(i) RNA (**16**) and DNA (**21**) nucleosides (See Fig. 2c(iii), shown below):

(ii) Other non-canonical nucleosides (**20**, **22–25**) and nucleotides (**26–31**) (See Extended Data Figure 1b and 1c, shown below):

The poor reactivity of aminoacyl-thiols (**1**) with the (primary) 5'-alcohol is shown by the reactions of 2',3'-dideoxyadenosine (compound **20A**) which only has a primary alcohol, compared to 5'-deoxyuridine (compound **23U**), which does not have a primary alcohol (only a diol) and undergoes

efficient aminoacylation (see Extended Data Figure 1c, above). This rules out intramolecular transfer from 5'-OH to 2',3'-diol.

The results with compound **22A** and **24A** which don't have a 3'-OH, and 3',5'-phosphate **29A**, alongside all our data with oligonucleotides (see Figure 5), also further rule out intramolecular transfer from 5'-OH to 2',3'-diol.

Our Supplementary Information further contradicts reviewer #2's assumptions with respect to thermodynamic stability. We observed that the small amount of 5'-aminoacylation (which is observed at higher pHs) is thermodynamically more stable than the 2',3'-aminoacylation. This can be seen clearly in Supplementary Figure 71 and is also shown below. This is the expected observation, if the selectivity is kinetically controlled due to the greater nucleophilicity of the 2',3'-diol.

Supplementary Figure 1. ¹H NMR yields (%) of aminoacylation on the 2'- or 3'-diol of **16U** (**17^U_{3'Ala}**, **17^U_{2'Ala}**, **17^U_{2'3'Ala}**, ■), and on the 5'-alcohol of **16U** (**17^U_{5'Ala}**, ○) from the reaction of **1^e_{Ala}** (200 mM) with **16U** (20 mM) using PET (15 mM) as an internal standard in A) MES buffer (1 M, pH 5.0); B) MES buffer (1 M, pH 6.5); C) MES buffer (1 M, pH 7.0); MOPS buffer (1 M, pH 7.5); MOPS buffer (1 M, pH 8). Set up following General Procedure C.

We have put forward an explanation for the observed selectivity in our paper, that the more reactive alcohols are those with lower pK_a and therefore the ones which are more easily deprotonated. Further speculation at this point would not be valuable – we have provided a model for the observed selectivity which fits all available data.

Comment 2J: *In order to judge the prebiotic relevance, it would be essential to learn about the stability of the 2',3'-ester again under “normal” pH-conditions. Richert et al observed that these esters have only a very limited stability, which is generally insufficient for efficient peptide formation reactions.*

Response 2J: We certainly agree that this is a very interesting question.

As we reported in our paper, the aminoacyl-RNAs are observed to be stable for days at pH 6.5 under our reaction conditions. They are less stable at higher pH, but under the optimal condition of our reaction they are stable for multiple days. See Figure 2c(ii) up to 20 hours, and Supplementary Figure 70 up to 100 hours.

The maximum yield of aminoacyl-RNA was equal at pH 6.0, 6.5 and 7.0. We observed that the aminoacyl-RNA were longest lived at pH 6.0 (see Supplementary Figure 70, which is also shown below), however the formation was slower to reach its maximum at pH 6.0 than pH 6.5 or pH 7.0.

We have reported our aminoacylation (predominately) at pH 6.5, as this balances rate of formation with longevity. It is also of note that we see higher 2',3'-diol > 5'-OH selectivity at pH 6.0 and 6.5, than at pH 7.0 (as we state in our paper) and so pH 6.5 was also selected to favour selective diol aminoacylation and therefore the natural position of tRNA aminoacylation.

Figure 2 c (ii) 2',3'-Aminoacyl-nucleoside **17** yield observed upon incubating alanine thioester L-**1**_{Ala}^e (200 mM) with adenosine **16A** (20 mM) at room temperature in MES (1 M, pH 5–6.5) or MOPS (1 M, pH 8) buffer. See Supplementary Figure 70 for pH 5–10 over 100 hours.

*Supplementary Figure 2. Total aminoacylation (%) of **16U** (20 mM) in the presence of **1**_{Ala}^e (200 mM) in 1 M buffer with PET (20 mM) as an internal standard. pH 5-6.5: 1 M MES buffer. pH 7-8: 1 M MOPS, pH 9-10: 1 M pyrophosphate. Set up following General Procedure C.*

Comment 2K: This is maybe the most critical issue. The reported methods for the activation of amino acids and the formation of the nucleoside esters of amino acids is only valuable if the formed activated structures can be used to form peptides. While this was shown by Richert and others, the manuscript of Powner ends with the formation of just the activated esters. If peptides can not be formed, the whole activation story would become a bit meaningless. I therefore recommend to investigate first if the activation chemistry allows the formation of peptides with

decent efficiency. The authors need to show that they can create peptides of a certain length with their chemistry before the importance of the thioester formation method and the observed chemoselectivities can be judged. In the current manuscript the activation chemistry is studied in great detail (without offering explanations for the observed chemoselectivities) but the value of the method has not been shown.

Response 2K: In response to this comment, we have **now demonstrated the robust, one-pot synthesis of peptidyl-RNA through the selective in-situ reaction of aminoacyl-RNAs**, under the conditions under which the aminoacyl-RNAs are formed (i.e., at pH 6.5), to afford peptidyl-RNA in near-quantitative yield.

Referee #2 raises an interesting question of where to draw a line under a particular investigation. They judge demonstration of peptide synthesis to be the key result to demonstrate the “value” of our method, we have now provided this requested peptide synthesis (see new Figure 6, also shown here below).

We cannot, of course, solve every outstanding question in prebiotic chemistry in one paper, but we agree that a complete narrative is essential. This is why we have endeavoured to perform a thorough investigation into RNA-aminoacylation by thioesters. Our results unexpectedly unite the biomolecules responsible for ribosomal and non-ribosomal peptide synthesis.

Biologically aminoacylation of RNA and peptide bond formation are separate processes catalysed by different ‘enzymes’. It is not reasonable for reviewer #2 to require that we simultaneously solve both problems non-enzymatically with one chemical method. Biology (life) did not do this; life split the process of protein synthesis into two separately controlled steps. Our aim was to learn about chemistry and selectivity that would be required to underpin the first step of this biological two-step process.

Beyond biochemistry, the referee highlights several reports including by Richert and coworkers [Jash *et al.* Single nucleotide translation without ribosomes. *Nat. Chem.* **13**, 751–757 (2021)] in which aminoacyl RNA is used as a substrate to form peptides in a purportedly prebiotically plausible manner. These reports further demonstrate the value of our current findings. These are the same compounds. And therefore, our aminoacylation neatly complements and fills an important hole in a (Richert’s) scheme from RNA to RNA controlled peptides – forming aminoacyl RNA under mild aqueous conditions. We have added a sentence to our manuscript to this effect, and included citations to the recent work by Richert *et al.* However, it is the natural (biological) process of RNA-aminoacylation, and the formation of the biological substrate that is required for protein synthesis that we are studying. There is no need to demonstrate aminoacyl-RNA can form peptides. There can be no question with respect to their biological value. Life on Earth could not exist without aminoacyl-RNAs, and **all proteins are biosynthesised from aminoacyl-RNAs**.

We have now also additionally demonstrated that peptidyl-RNA formation on aminoacyl-nucleosides is not only feasible but can be effective and highly selective in situ, under the same condition as RNA is aminoacylated selectively by aminoacyl-thiols (See Figure 6 in our manuscript, also shown above).

Reviewer #2 recommends that we need to: *investigate first if the activation chemistry allows the formation of peptides with decent efficiency to demonstrate the value of our reaction.*

We now have demonstrated that the reaction of aminoacyl-nucleosides **17**, formed in situ from aminoacyl-thiol (**1**) and nucleoside/nucleotide, with (prebiotic) peptide-thioacids (e.g. Ac-Aaa-SH) yields peptidyl-RNA in near-quantitative yield in a highly chemo-selective one-pot reaction at pH 6.5. We believe this is not only “*decent efficiency*”, but the most effective chemical synthesis of peptidyl-RNA reported in the literature. See Figure 6 in our manuscript (shown above) and below data (or Extended Data Figure 8):

Entry	aminoacyl-RNA		peptide-thioacid 11		40
	#B	Aaa	Aaa	(mM)	(%)
1	17A	Arg	Gly	(105)	93
2	17G	Arg	Gly	(150)	92
3	17U	Arg	Gly	(200)	97
4	17C	Arg	Gly	(200)	96
5	17A	Ala	Gly	(300)	90
6	17A	Glu	Gly	(300)	92
7	17A	Leu	Gly	(350)	95
8	17A	Lys	Gly	(300)	93*
9	17A	Ser	Gly	(300)	91
10	17A	Arg	Val	(300)	88

Entry	aminoacyl-RNA		peptide-thioacid 11		40 or 42
	#B	Aaa	Aaa	(mM)	(%)
11	17A	Arg	Phe	(300)	85
12	17A	Arg	Met	(300)	87
13	17A	Arg	GlyGly	(110)	92
14	17A	Arg	AlaAla	(150)	89
15	17A	Arg	AlaPro	(200)	82
16	17A	Arg	MetGly	(120)	77
17	17A	Arg	GlyGlyGly	(200)	96
18	41A	Arg	Gly	(150)	92
19	41A	Arg	GlyGly	(110)	93
20	41A	Arg	GlyGlyGly	(100)	95

These experiments perfectly highlight the surprising and subtle differences between two superficially similar electrophiles derived from amino acids – and importantly the ability to **chemically control aminoacylation and peptide formation in water at the same pH**.

Aminoacyl-thiols (**1**) react selectively with RNA-diols but not amines in water at pH 6.5, while activated peptide-thioacids (**11**) react selectively with amines but not diols *under the same conditions in terms of pH and concentration!* This is a remarkable observation, and we do not know of another chemical reaction that can achieve this type of switch in reactivity under the same (pH, concentration) conditions.

We thank the referee #2 for their suggestion which has prompted this additional work, we think **these new experiments are an elegant demonstration of the key point of our paper** and the remarkable reactivity of (biological) aminoacyl-thiols (**1**).

To our knowledge this new addition (peptidyl-RNA synthesis) is the only high-yielding, protecting-group-free, chemically controlled, one-pot synthesis of peptidyl-RNA (**40**) in water that has been reported in the literature.

Comment 2L: *Finally, I find the manuscript is overloaded with data to the different amino acids, which makes it hard to discover the key points. This over-complexity (particularly of the Figures) somehow blocks the ability to understand the key messages.*

Response 2L: We find this comment, that we present “*too much data*”, to be an unusual complaint, particularly in light of the referee’s request for us to expand the scope of our investigation to include peptide formation!

We have now demonstrated one-pot peptidyl-RNA synthesis, this of course added significant work and additional data to our paper, but we are grateful to reviewer #2 for this recommendation

as these new results highlight how truly remarkable the chemistry of aminoacyl-thiols is to be able to control the protecting-group-free synthesis of aminoacyl-RNAs.

We believe that our study of the full range of canonical amino acid side chains is an important aspect of our work. This is an aspect of prebiotic peptide chemistry that is all too often overlooked! Without the breadth of our study, we would not have found that Arg possesses an intramolecular catalytic motif that accelerates aminoacylation or shown that Asp has exquisite kinetic selectivity for the canonical alpha-isomer – these are extremely important and unprecedented results.

To accommodate reviewer #2's concern, however, we have now separated out the studies of amino acid sidechains into a separate section of the revise manuscript, so these novel findings with respect to sidechain reactivity, can be appreciated separately and do not cloud the other messages within the paper. We have reported many unprecedented and chemically important findings; this should not be a barrier to publication.

Overall, we are grateful to the referee for engaging thoughtfully with our work, and hope that the changes we have made will reassure them that the results which they find so surprising will also be of broad interest to the wider scientific community.

Point-by-point response to reviewer's comments

We would like to thank all our reviewers for their constructive comments and positive appraisal of our work.

Referee #1:

Comment 1a: The authors have substantially revised and improved their paper. The selectivity for aminoacylation vs peptide formation is remarkable and is an important finding. I recommend publication of this paper.

Response 1a: We thank reviewer #1 for their comments and positive appraisal of our work.

Comment 1b: My one remaining reservation concerns the statements in the abstract about the selectivity of the aminoacylation reaction. While the aminoacylation chemistry is certainly selective for the cis-diol of RNA vs. over amines, it is not selective over internal 2'-hydroxyls in ss RNA (as shown in lines 216-219). The current phrasing on diol selectivity does not make this clear. It would therefore be better to shorten the abstract as follows: From...aminoacyl-thiols (**1**) react selectively to yield 2',3'-aminoacyl-RNA at neutral pH. **1** react selectively with RNA-diols over amine nucleophiles, promoting aminoacylation over adventitious (non-coded) peptide bond formation. To ... aminoacyl-thiols (**1**) react selectively with RNA-diols over amine nucleophiles, promoting aminoacylation over adventitious (non-coded) peptide bond formation.

Response 1b: Change made exactly as specified by reviewer #1.

However, we note that Watson-Crick base pairing blocks internal 2'-hydroxyl aminoacylation (as shown in Figure 3 in our manuscript) and so aminoacylation is highly selective for the terminal 2',3'-diol, over internal 2'-hydroxyls, in ds-RNA.

Comment 1c: The following minor comments (mostly typos) should also be addressed.

Response 1c: We thank reviewer #1 for highlighting these.

Comment 1d: Line 112: typo, should be 'despite'

Response 1d: Corrected.

Comment 1e: Line 298: what ribozymes?

Response 1e: The ribozyme is specified in the next sentence (on line 299), where it is stated: "... Specifically, an RNA-duplex (**ON10** + **ON13**) with a mis-paired 3'-U/5'-U and 5'-G overhang,¹⁵ amplified 3'-aminoacylation yields (Fig. 4c) ...".

Comment 1f: Line 454: with peptide-thioester Ac-Ala-Set is not a peptide thioester, it is just an N-acetyl-aminoacyl ester

Response 1f: Changed to "*N*-acetyl-aminoacyl-thiol".

Comment 1f: Line 469: underling should be underlying

Response 1f: Corrected.

Comment 1g: Line 546: Should be orchestrated

Response 1g: Corrected.

Comment 1h: Lines 573 and 575, should be adenylate

Response 1h: Corrected.

Referee #4:

Comment 4a: This is a remarkable manuscript. Singh et al. demonstrate the selective aminoacylation of nucleoside and oligonucleotide in water with thioester, and the synthesis of elongated peptidyl-RNA from one-pot reactions between the resulting 2',3'-aminoacyl-riboside with peptide-thioacid. The work strikes the right balance in selective activation chemistry to properly funnel reactants down paths that lead to the synthesis of peptides in a ribonucleoside dependent fashion, as found in extant biology. Along the way, the authors demonstrate how thioesters can be generated from N-carboxyanhydrides and thiols, how duplexes of RNA can guide aminoacylation to the terminus of RNA, and how a terminus similar to that of tRNA facilitates aminoacylation. The work is quite thorough and could be split into multiple papers, although I would not advice doing so. In the current manuscript, a logical, easy to follow, complete story is presented that will, in my opinion, make a strong, long-lasting impact on the prebiotic chemistry field.

I have gone through the comments from review 2 and find all of the responses thorough and convincing. The authors made appropriate changes to the manuscript and have now even demonstrated coupled peptide bond formation.

I have no interest in suggesting more experiments. There is already more than enough there to make a convincing story (and the manuscript already has 284 pages of supplementary information).

Response 4a: We thank reviewer #4 for their thorough appraisal of our work and prior responses. We are especially grateful for their positive engagement with respect to suggesting future investigations beyond the scope of this paper.

Comment 4b: I do have a few comments that the authors may wish to consider, but again, I am NOT requesting more experiments. The authors note that the thioester of Arg was particularly effective in aminoacylation and attribute this activity to the formation of a cyclic intermediate. That may very well be the case, but the known ability of Arg to interact quite effectively with RNA seems to be overlooked. Is it not reasonable to expect that a molecule that has greater intrinsic ability to interact with RNA would also aminoacylate RNA more effectively? RNA-binding proteins are typically enriched in Arg, and Lys does not interact as extensively with RNA as Arg.

Response 4b: This is an interesting point, and one which we had considered before observing the formation of cyclic arginine **33** during aminoacylations. Following in-situ observation of **33**, we isolated this intermediate and demonstrated its enhanced rate of aminoacylation, compared to the thioester **1Arg**, and so its catalytic effect (see Supplementary Table 13). It is in principle possible that there are 'two contributing factors' that augment Arg reactivity, but this is not the simplest conclusion to draw from the data and our observation. Furthermore, Arg enhanced aminoacylation is observed with (neutral, unphosphorylated) nucleosides (e.g., **16A** and **16U**), which demonstrates the Arg-effect is not due to either phosphate-guanidinium hydrogen bonding or columbic bonding (electrostatic) interaction, and whilst hydrogen bonding or π -stacking interaction with the nucleobase might potentially be proposed to play a role, this is not a reasonable explanation for our observations given our results with **16A** and **16U** (and their markedly different chemical structures). Our current model for sidechain catalysis explains the data that we have observed.

However, we strongly agree with reviewer #4 that the 'effective' sidechain specific interactions, that have been previously observed, between Arg residues and RNAs are very interesting. This observed supramolecular affinity, together with the highly effective nature of RNA-arginylation we have now observed, warrants further investigation of these interactions in the context of structure and function of arginylated RNAs. Therefore, in response to reviewer #4 requested, we have now noted this in our manuscript (highlighted in yellow, on Line 286). We have not observed these Arg/RNA binding interactions in our data, with all the sequence of RNA tested aminoacylating with arginine comparably (see Supplementary Figure 112) and so we have added additional citation for where the interactions have been previously observed.

Comment 4d: I'd also like to note that the guanidine/guanidinium group of Arg is not shown protonated in figure 4, and I'd expect that the protonated form would dominate at the pH of the experiments. A justification for this in the text seems lacking.

Response 4c: This is correct. The Arg sidechain (pK_{aH} 13.8) will be almost completely protonated at pH 6.0–7.0. We have now noted in Figure 4 legend that: "The Arg sidechain (pK_{aH} 13.8) is drawn neutral but will be protonated at pH 6.0–7.0."

The chemical structures are uniformly shown in their neutral form. Many of the structures shown are employed in reactions near (one of) their pK_a 's (e.g., key compound **1**, pK_{aH} = 7.5) and so both neutral and charged forms are present in these reactions at a pH dependent ratio, and other compounds have multiple sites of potential protonation/deprotonation. Depicting this is (for each reaction) would be an unnecessary complication.

We had specified in the original manuscript that "Structures are drawn neutral for clarity" but this was lost during our extensive revision of the figures in this manuscript. We have now added this note back into the start of the manuscript, in the legend for Figure 1.

Comment 4d: Am I understanding correctly that the ability of Cys to react with amino acid thioester to form a dipeptide is not problematic in the scheme proposed by the authors? Wouldn't the resulting dipeptide be capable of further extension with thioacid, as the Cys would no longer reside at the amino-terminus? That is, extension could continue with the residue at the N-terminus and the now more internal Cys would not continue to appreciably react with thioesters? If that's the case, perhaps that's worth mentioning.

Response 4c: This is correct. This is not problematic in our scheme. *N*-Acyl-cysteines are thiols, and we have shown that they function as competent aminoacyl-donors in our aminoacylation reactions (see Supplementary Table 10).

And yes, the resultant cysteine peptides are capable of extension with thioacids; we demonstrated cysteines undergo highly effectively ligation with thioacids in Canavelli+ *Nature* **2019**, 571, 546–549, even a (free) *N*-terminal cysteine amino acid. We have already noted this in the manuscript on lines 501-503.

Comment 4d: There are nucleotides, such as NAD(H), that are important for metabolism. I wonder if thioesters would efficiently aminoacylate such nucleotides, and what the effect of such activity would be. Again, I'm not requesting that this experiment be run or even that the authors comment on this in the manuscript. This is more a comment to consider (or not) for future studies.

Response 4c: This is an excellent suggestion for our future work. We will study the effects of aminoacylation upon the reactivity and function of RNA-cofactors, but these studies are beyond the scope of this current paper.

Comment 4d: The authors suggest "(arctic or polar) soda-lakes" as a potential geological setting. I know that it is difficult to say with much confidence where the described chemistry could have taken place, but I think it is useful for the field to actively think of plausible settings. As the proposed chemistry exploits many sulfur-containing molecules (and also exploited iron as $K_3Fe(CN)_6$), there are clearly more constraints on what would have been necessary. There are also many other papers from the Powner group and colleagues that are related and necessary for the described scenario. What do all of these data tell us regarding a plausible setting?

Response 4c: We agree with reviewer #4, whilst consideration of plausible environments should, in our opinion, in no way limit research into the chemical mechanisms that may have underpinned the origins of life, it may be useful or productive for researchers in the field to actively consider plausible settings. We believe this to be especially true for any reactions environments or conditions that are non-trivial to rationalise. This is precisely why we have very carefully considered how, under prebiotically plausible conditions, a naturally neutral solution could become (reversibly) acidified to pH 3–5. We realised that this pH transition could be readily, simply and plausibly achieved (based on literature precedent, see reference 41) upon freezing a neutral pH phosphate solution. We then successfully demonstrated that freezing a neutral pH phosphate solution, and the resultant temporary acidification, could be applied to the effective synthesis of aminoacyl-thiol **1** from aminonitriles (as shown in Figure 5d and Supplementary Figure 195).

We have suggested that (arctic or polar) soda-lakes may have played a role in this freezing process because the highest concentration natural phosphate solution observed on the Earth today are observed in soda lakes, and, of course, arctic and polar lakes often experience subzero (freezing) temperatures (see reference 44). Further investigation and discussion of geochemical settings is beyond the scope of this paper.

Comment 4d: I have to say that I have grown tired of reading exciting titles that then let me down when I read the actual work described in the paper. I felt the opposite with this manuscript. The title was intriguing, of course, but my enjoyment only increased as I continued to read the paper. It's very nice work.

Response 4c: We thank reviewer #4 for these comments, and again for their insightful thoughts and suggestions for our future investigations.